# Orthogonal luminescence lifetime encoding by intermetallic energy transfer in heterometallic rare-earth MOFs

Jacob I. Deneff[1], Lauren E. S. Rohwer[2], Kimberly S. Butler [3], Bryan Kaehr [4], Dayton J. Vogel [5], Ting S. Luk[6], Raphael A. Reyes[1], Alvaro A. Cruz-Cabrera[7], James E. Martin[8] & Dorina F. Sava Gallis[1] ✉

Lifetime-encoded materials are particularly attractive as optical tags, however examples are rare and hindered in practical application by complex interrogation methods. Here, we demonstrate a design strategy towards multiplexed, lifetime-encoded tags via engineering intermetallic energy transfer in a family of heterometallic rare-earth metal-organic frameworks (MOFs). The MOFs are derived from a combination of a high-energy donor (Eu), a low-energy acceptor (Yb) and an optically inactive ion (Gd) with the 1,2,4,5 tetrakis(4-carboxyphenyl) benzene (TCPB) organic linker. Precise manipulation of the luminescence decay dynamics over a wide microsecond regime is achieved via control over metal distribution in these systems. Demonstration of this platform's relevance as a tag is attained via a dynamic double encoding method that uses the braille alphabet, and by incorporation into photocurable inks patterned on glass and interrogated via digital high-speed imaging. This study reveals true orthogonality in encoding using independently variable lifetime and composition, and highlights the utility of this design strategy, combining facile synthesis and interrogation with complex optical properties.

As markets become more complex and globalized, more sophisticated methods are required to address supply chain management challenges and integrity-checking for highly complex systems and valuable items or commodities. To meet this growing challenge, optical tags, which leverage the luminescent properties of materials for encoding, have been explored and advanced significantly in recent years[1–3]. To be effective, such tags must offer rapid, low-cost, and unambiguous verification, and, crucially, they must be secure against counterfeiting. Because static, monochromatic luminescence is easy to duplicate using commonly available dyes, multilayered encoding must become

the state of the art, reducing the likelihood that any counterfeit will duplicate all of the measured properties[4].

To enhance security, viable materials must combine many possible states, both overt and covert, including emission, lifetime, scattering, and absorption[5]. In agreement with this premise, several recently published works have described potential options for secure encoding[1,6–9], drawn from many classes of materials, including carbon dots[2,10], metallic nanoparticles[11,12], and perovskites[3]. However, existing materials suffer from critical drawbacks: (i) monochromatic and/or broad emission bands, unsuited for multi-tiered screening[3,13];

[1]Nanoscale Sciences Department, Sandia National Laboratories, Albuquerque, NM 87185, USA. [2]Advanced Packaging/Integration Department, Sandia National Laboratories, Albuquerque, NM 87185, USA. [3]Molecular and Microbiology Department, Sandia National Laboratories, Albuquerque, NM 87185, USA. [4]Advanced Materials Laboratory, Sandia National Laboratories, Albuquerque, NM 87185, USA. [5]Computation Materials and Data Science, Sandia National Laboratories, Albuquerque, NM 87185, USA. [6]Nanostructure Physics Department, Sandia National Laboratories, Albuquerque, NM 87185, USA. [7]Measurement Science and Engineering Department, Sandia National Laboratories, Albuquerque, NM 87185, USA. [8]Nuclear Security Engineering Department, Sandia National Laboratories, Albuquerque, NM 87185, USA. ✉e-mail: dfsava@sandia.gov

(ii) complex synthesis methods, which are time-consuming and cost-ineffective[11,14]; and (iii) highly specialized interrogation equipment, which restricts real-life implementation[2,15].

Here we demonstrate that polynuclear rare-earth metal-organic frameworks (MOFs) represent an ideal class of materials for tunable, multiplexed encoding[16–22]. Rare-earth elements (REs) possess narrow absorption and emission bands across a wide range of wavelengths that allow for a high degree of specificity in excitation and emission[23], a property that is difficult to duplicate with the broad-band materials[10]. Likewise, MOFs are attractive for tagging because they have (i) straightforward synthetic processes, that allow for one-pot synthesis; (ii) periodic crystal structures, allowing structural manipulation and in-depth characterization at the atomic level; and (iii) sensitizing organic linkers in a low-density framework, which only require a low amount of optically active REs to emit a readable signal[24,25].

Lifetime-encoded materials are particularly attractive for multiplexed optical tags. Distinct luminescence lifetimes allow spectrally overlapped tags to be distinguished in the time domain[2,3,26], and can create images that change over time to reveal hidden patterns or additional data. For example, organic compounds with variable lifetime have been reported, including thermally activated delayed fluorescence materials[2,27]. However, the challenges associated with controlling the material structure, have limited the rational design of materials with potential for lifetime-encoding[2,3,8,11,28,29].

Available energy transfer pathways in a tag are affected by both the excited state energies of individual components of the material and the distance between them. The periodic structure of polynuclear rare-earth MOFs, with intermetallic distances controlled by cluster and ligand geometry, makes the energy transfer relationships between the various components of the rare-earth MOF much easier to understand and manipulate[30,31]. Manipulation of MOFs for control of emission color or intensity has been well explored, but the concept has never been adapted for fine control of luminescence lifetime[30–32]. When changes in lifetime are noted, both in MOFs and other lanthanide-based materials,

they are connected to the ratio of two elements in the material, directly coupling composition and lifetime. To extend encoding potential, here we report an optical tag that demonstrates orthogonal encoding using independently variable lifetime and composition.

To enable this multiplexed encoding, we implemented a targeted materials design strategy: first, the dominant metal-metal interactions were confined to intra- cluster vs. inter-cluster interactions via linker geometry, which allowed the control over the intermetallic energy transfer mechanism; then, the high-energy donor RE and lower-energy acceptor RE were segregated by incorporating an optically inactive RE to modulate the average distance between the donor and acceptor without changing the linker or overall structure of the MOF[30].

The resulting family of heterometallic rare-earth MOFs combined the high-energy donor Eu (visible emitter) with the low-energy acceptor Yb (near-infrared (NIR) emitter) and the optically inactive Gd ion with the organic linker 1,2,4,5 tetrakis(4-carboxyphenyl) benzene (TCPB). The TCPB linker was chosen due to its proven ability to direct nonanuclear cluster formation in the MOF and because its large size allows for cluster separation[17,33]. Due to the energy of its triplet excited state, TCPB also serves as antenna molecule for both Eu and Yb, but not Gd[16,34]. We hypothesized that direct excitation of the linker initiates an energy cascade first to Eu (which has a long-lived visible emission), then to Yb (which has a short-lived NIR emission), while avoiding any energy transfer to Gd, Fig. 1. As a result, here we demonstrate that the inclusion of an inactive element vastly expands the potential design space for lifetime encoding, because it effectively modifies the RE interactions without impacting either the linker identity or the ratio of donor to acceptor emitters.

To validate the complex intermetallic energy transfer relationships between these three elements, we targeted and successfully synthesized 13 compositions. These compositions included single-, di-, and trimetallic compounds with different elemental ratios, which allowed us to explore a wide range of relevant intermetallic energy transfer scenarios.

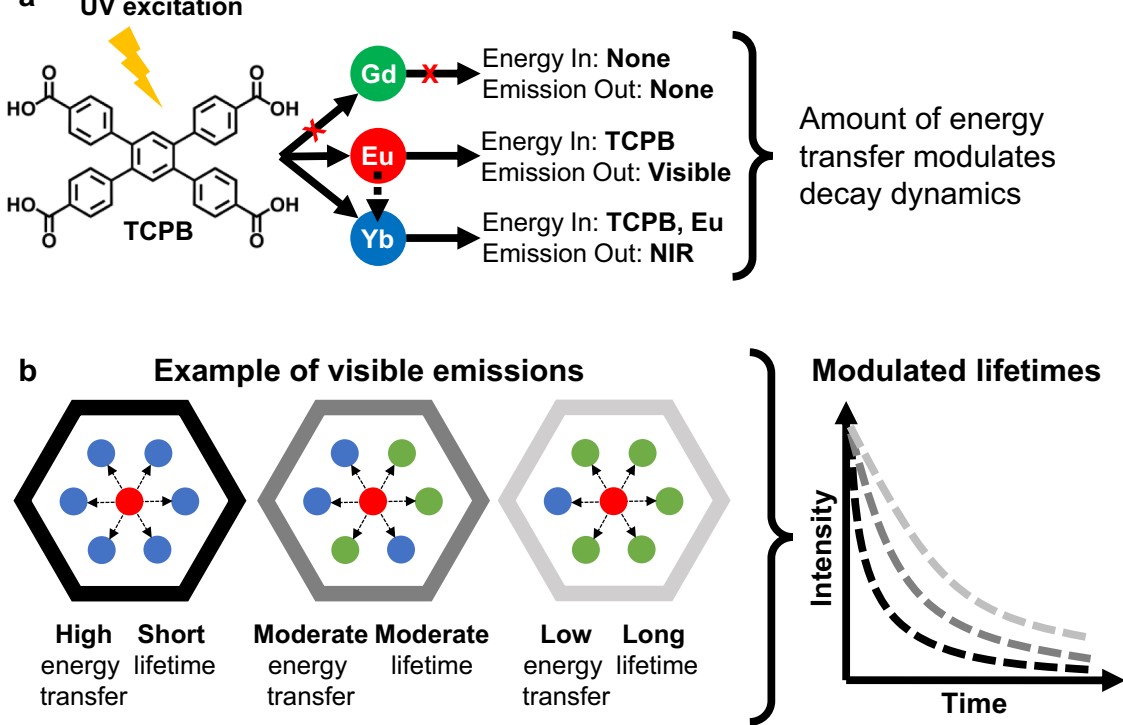

Fig. 1 | **Concept of controllable energy transfer for lifetime modulation in EuGdYb-based trimetallic compositions. a** Direct ligand excitation results in modulated energy transfer, illustrating the relationships between each component. **b** Examples of complex intermetallic energy transfer in three distinct EuGdYb-based compositions with different elemental ratios and the direct impact on lifetime modulation.

To demonstrate the utility of these materials as practical tags we implemented two design strategies, leveraging two spectrally identical compositions that displayed substantially distinct decay lifetimes in the 100 s of µs. An encoding strategy that utilizes the braille alphabet was introduced, interrogated by a microplate reader to produce 3 tag exemplars that are either static or dynamic, utilizing both concentration and lifetime to change the encoded message depending on when the pattern is read. Subsequently, a proof-of-concept tag was fabricated utilizing the same two compounds suspended in photocurable adhesive and patterned with a laser cut stencil. Laser excitation and high-speed imaging interrogation of this tag showed that the two materials were clearly visually distinguishable on a millisecond time scale. These demonstrations show the utility of these tag materials in facile encoding and interrogation, while more in-depth characterization of a library of tag materials reveals the potential for further layers

of encoding including additional lifetime lengths, compositions, and luminescence spectra.

The materials and techniques presented here demonstrate a versatile materials platform capable of dynamic encoding in the time domain and amenable to truly independent authentication via tunable emission spectra and composition, enabling the future creation of accessible tags with multi-layered encoding.

## Results and discussion
### Synthesis and materials characterization
The rare-earth MOFs in this study contained one, two, or three metals, and were chosen to fill the composition space of the three metal ions (Eu, Yb, and Gd) in consistent increments, depending on the number and ratio of metals incorporated. Figure 2a shows a ternary composition diagram with the specific compositions explored in this work. The

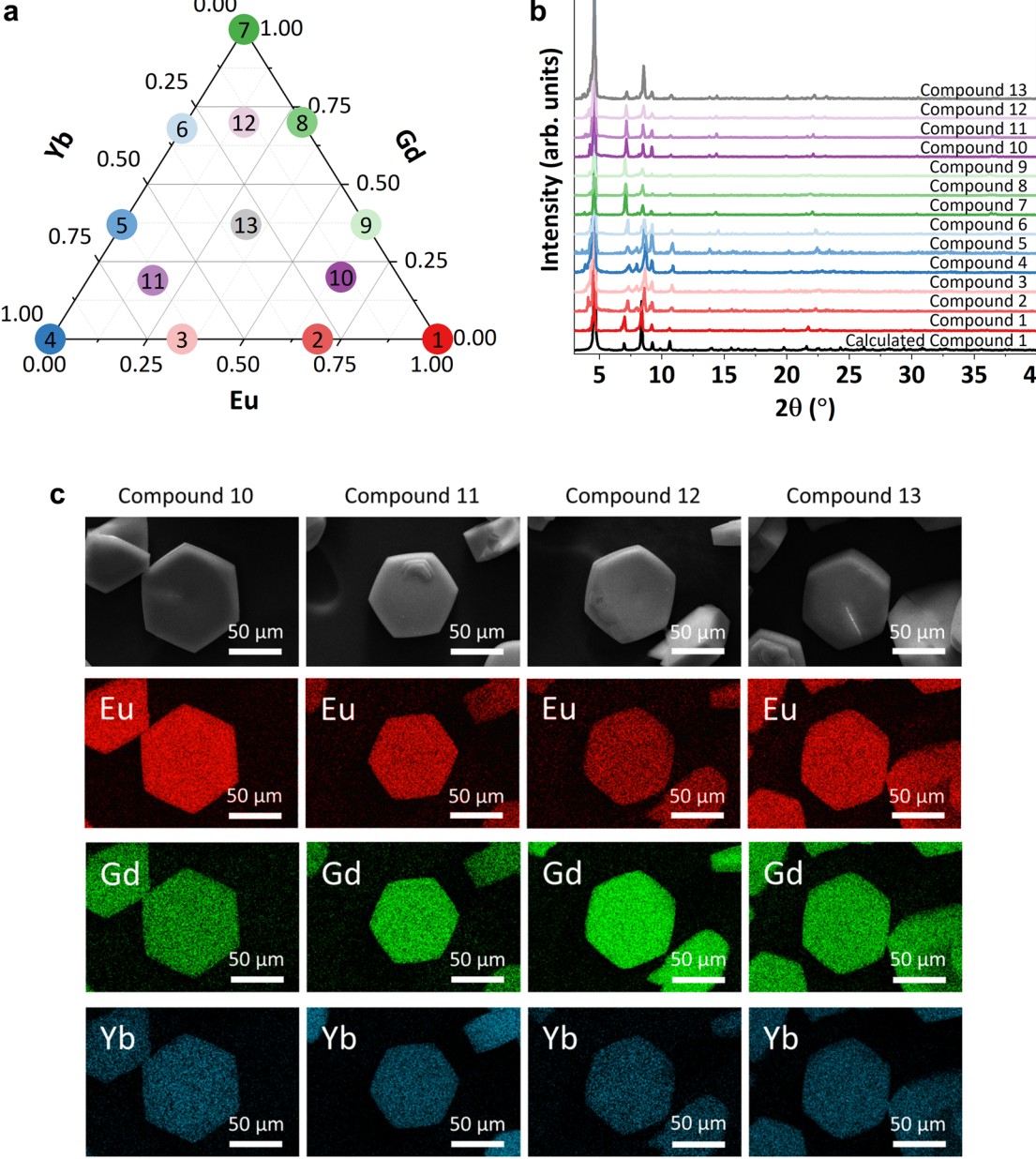

**Fig. 2 | Information regarding the composition and crystallinity of the reported compounds. a** Ternary diagram of metal content showing the composition of each compound in the study. **b** Powder X-ray diffraction patterns for each compound reported in this work, highlighting their consistent crystallinity and the slight variations resulting from different metallic compositions. **c** Scanning electron microscope images and energy dispersive spectroscopy maps for the four trimetallic compounds **10**–**13**, highlighting the morphology of the materials and the uniform distribution of metals within each crystal.

**Table 1 | Compound ID numbers and their fractional metallic compositions**

| Compound | Eu fraction | Gd fraction | Yb fraction |
|---|---|---|---|
| 1 | 1 | 0 | 0 |
| 2 | 0.69 | 0 | 0.31 |
| 3 | 0.34 | 0 | 0.66 |
| 4 | 0 | 0 | 1 |
| 5 | 0 | 0.37 | 0.63 |
| 6 | 0 | 0.68 | 0.32 |
| 7 | 0 | 1 | 0 |
| 8 | 0.30 | 0.70 | 0 |
| 9 | 0.63 | 0.37 | 0 |
| 10 | 0.65 | 0.20 | 0.15 |
| 11 | 0.17 | 0.19 | 0.64 |
| 12 | 0.15 | 0.70 | 0.15 |
| 13 | 0.32 | 0.37 | 0.31 |

The reported fractional compositions have an uncertainty of +/−0.01 based on EDS analysis.

representative crystal structure for these rare-earth MOFs was previously detailed elsewhere by us and others[17,33]. Briefly, the three-periodic structure is derived from nonanuclear clusters linked by 12 carboxylate groups of the TCPB linker. The resulting framework possesses intrinsic porosity accessible via 1D channels of ~1.2 nm. Crystal structure data and an illustration of compound **1** can be seen in Supplementary Table 1 and Supplementary Fig. 1, respectively.

The crystallinity and phase identity of each compound was determined via powder X-ray diffraction (PXRD). Figure 2b shows the patterns for each compound, along with a calculated pattern for the homometallic Eu composition (compound **1**) for comparison. The results indicate that each sample is phase pure and closely matches the representative calculated pattern for the Eu compound **1**. Minor shifts in peak location between different compounds are reflective of the different ionic radii of the REs. Notably, these materials possess a high thermal stability up to 500 °C, as determined by thermogravimetric analyses, Supplementary Fig. 2.

The compositions of each di- and trimetal compound were determined via scanning electron microscopy-energy dispersive spectroscopy (SEM-EDS) imaging and analysis. The metallic composition of each compound is given numerically in Table 1 and visually as a ternary diagram in Fig. 2a. Microscopy images and the corresponding elemental maps for the trimetallic compositions (compounds **10**, **11**, **12**, and **13**) are shown in Fig. 2c. The SEM-EDS analyses for all other heterometallic compositions are included in Supplementary Fig. 3. EDS mapping shows an even distribution of each metal, both within individual crystals and in the bulk sample.

This finding is significant, as the synthesis of heterometallic MOFs is non-trivial. Heterometallic MOFs rely on the compatible coordination geometries and crystallization kinetics of these metals to avoid phase or domain separation[35,36]. Because REs all possess similar coordination chemistries, sizes, and oxidation states, it is assumed that rare-earth MOFs can be constructed without domain separation and without limits on the relative metallic composition[23,32,37,38]. Nevertheless, to address concerns regarding uniformity within the heterometallic compositions described here and ensure that our compounds were uniform, we analyzed nine points on a single crystal and three areas on different crystals of compound **13**. The results, displayed in Supplementary Fig. 4, indicate that the distribution of elements is indeed uniform both in individual crystals and across the bulk sample.

## Intermetallic energy transfer

Much of the complexity and tunability of heterometallic polynuclear cluster-based rare-earth MOFs is derived from energetic interactions between their components, primarily ligand-to-metal charge transfer (LMCT) and metal-to-metal charge transfer (MMCT)[24,39]. The efficiency of these pathways is governed by both the relative energies of each component and the distance between them. Components with high-energy excited states will donate energy non-radiatively to components with low-energy excited states, provided that the elements are in close physical proximity to each other and that the energy levels are sufficiently different to prevent losses via back-transfer[4,40].

The prototypical three-periodic MOF material studied here is made up of 12-connected nonanuclear RE clusters bridged by TCPB linkers[17,18,33]. Within individual clusters (intra-cluster) the intermetallic distance is 3.9 Å, while distances between clusters (inter-cluster) are between 8.9 and 18.0 Å, as shown in Fig. 3a.

Given that intermetallic energy transfer is known to decline significantly beyond 10 Å[31], these distances indicate that energy transfer between individual metal ions in the compositions reported here would primarily occur within clusters. This paradigm is significant in the context of correlating metal distribution at the molecular level and the impact of such distribution on the metal-to-metal energy transfer and resulting photophysical properties. To help explain the photoluminescence (PL) response of each compound to the direct excitation of the linker, the energy transfer pathways between the different components of the MOFs are shown in Fig. 3b.

Further evidence for the dominance of intra-cluster energy transfer is found in the PL spectra for the reported compounds (Fig. 3c, d). For example, when Eu is excited directly with 394 nm light, characteristic Eu emission peaks in the visible range are observed ($^5D_0 \rightarrow {}^4F_0$ at 582 nm, $^5D_0 \rightarrow {}^4F_1$ at 592-600 nm, $^5D_0 \rightarrow {}^4F_2$ at 617 nm, $^5D_0 \rightarrow {}^4F_3$ at 655 nm, and $^5D_0 \rightarrow {}^4F_4$ at 704 nm)[41]. Critically, Yb emission in the NIR is observed between 970 nm and 1050 nm under the same excitation wavelength, demonstrating Eu-to-Yb energy transfer[41]. As expected, Gd does not affect the PL spectra of materials that contain it, due to its high-energy excited state. Additional excitation and emission spectra for the TCPB linker and single metal compounds **1**, **4**, and **7** are presented in Supplementary Fig. 5.

The environment surrounding some RE ions is known to affect their luminescence; for example, the site symmetry of the Eu is indicated by the splitting of its $^5D_0 \rightarrow {}^4F_1$ transition emission peak[41]. The Yb signal is also typically split into several peaks in the 970 to 1050 nm range, which reflects the local concentration of Yb and other elements in the material[42]. Thus, all of the compounds in Fig. 3d show primary Yb emission peaks located at 975 and 982 nm, which change in relative intensity primarily in response to the presence or absence of Eu in the compound. This phenomenon of changing relative intensities reflects the local environment of the Yb ions and would not occur unless the metals were mixed within the same cluster and interacting in that space.

Finally, we use infrared (IR) transmission spectroscopy to probe the vibrational modes around metal clusters within the different compounds (results shown in Supplementary Fig. 6). The vibrational modes associated with bridging the OH groups in the MOF metal clusters can be found in the 800–1000 cm$^{-1}$ range[43], and changes in the ratios of available metals have been shown to cause peaks to shift or emerge to reflect the different environments[44]. Because of the similar coordination chemistry of the different REs, we did not expect new peaks to form in response to changing composition; however, we did observe peak shifts in the homometallic compounds (particularly compounds **1** and **4**), primarily around 910 cm$^{-1}$ and 845 cm$^{-1}$. Mixed metal compounds show intermediate peak shifts that correspond to the relative quantities of Yb and Eu. Peaks shift rather than split due to the presence of OH groups, which bridge both single metals and mixed metals. While not conclusive in and of themselves, these results further support our above stated hypothesis of intra-cluster heterometallic mixing rather than distinct homometallic clusters.

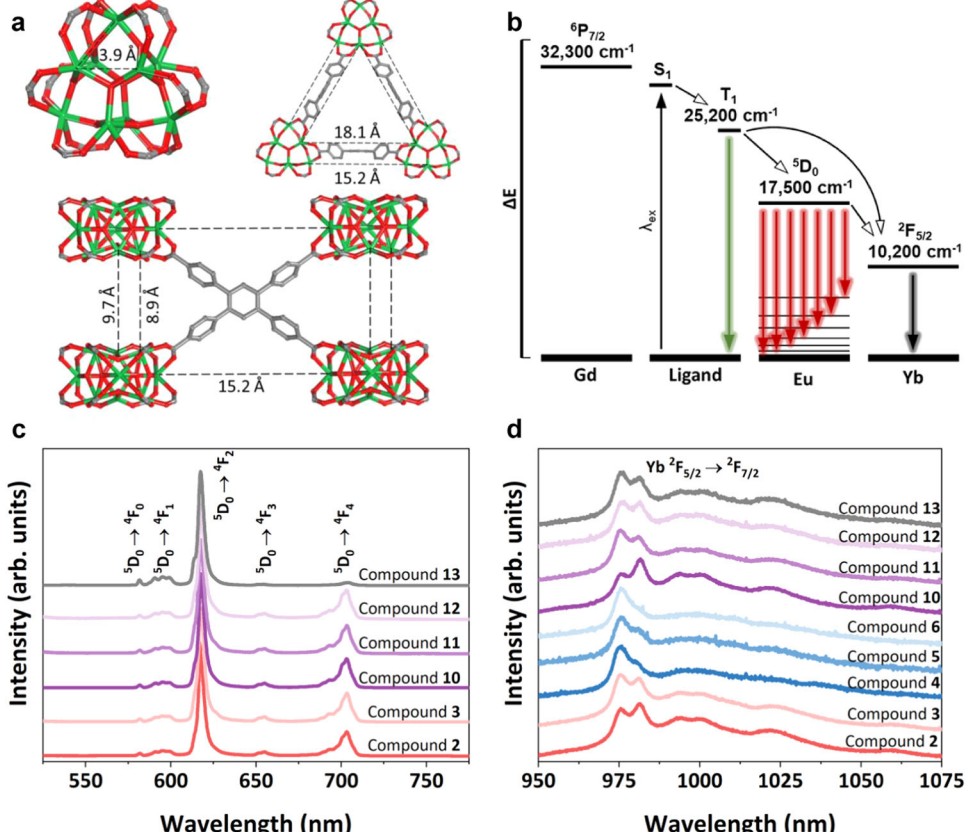

**Fig. 3 | Detailed MOF structural information and photophysical properties evaluation of the rare-earth MOF family. a** Ball-and-stick representation of MOF structure showing the intra-cluster and inter-cluster intermetallic distances in compound **1**, EuTCPB. Atom color scheme: C gray, O red, Yb green. H atoms are omitted for clarity. **b** Energy transfer diagram illustrating the relationships between each component of the compounds reported here when the ligand is excited directly ($\lambda\_ex$ = 337 nm). Colored arrows were chosen to approximate emission color of each transition. **c** Emission spectra for select compounds in the visible range, with the excitation wavelength 394 nm; all peaks associated with Eu. **d** Emission spectra for select compounds in the NIR range, with the excitation wavelength 394 nm; all peaks associated with Yb.

## Photoluminescent lifetime measurements

To probe the entire complex energy transfer cascade from ligand to metal and metal to metal, we measured the luminescence lifetime of each emitting compound. Both visible and NIR decay dynamics were measured using 337 nm light to target direct linker excitation. Based on its emission spectrum, the donor state energy of TCPB is ~25,200 cm$^{-1}$[16,32], while the primary emissive states of Gd, Eu, and Yb are 32,300, 17,500, and 10,200 cm$^{-1}$ respectively[41]. Because of their relative energy levels, a cascade of energy transfer interactions were anticipated: Eu was excited by energy transfer from the linker, and Yb was excited by energy transfer from both the linker and neighboring Eu ions.

Photoluminescent decay in the visible range in compounds **1–3** and **8–13** were measured at 620 nm to target the $^5D_0 \rightarrow {}^4F_2$ transition in Eu and fitted by a single exponential function (Fig. 4a–c). The characteristic decay times for these fitting functions are given in Table 2. Figure 4a shows data for compounds **1**, **8**, and **9**, and demonstrates that diluting Eu with the inactive metal Gd has little effect on the lifetime of the material, because the characteristic decay time is ~400 μs regardless of composition. Figure 4b shows that, when Yb is included with Eu in compounds **2** and **3**, the energy from a long-lived emitter (Eu) drains to a short-lived one (Yb), resulting in an overall decrease in the visible lifetime from 367 μs for the pure Eu compound **1** to 125 μs for the Yb rich compound **3**.

After exploring the binary interactions between metals in the MOFs, we studied trimetallic compositions, combining Gd, Eu, and Yb into a single compound and producing more complex behavior.

Figure 4c shows the lifetime decay dynamics for the trimetallic compounds **10–13**. The lifetimes for these compounds range from 122 to 300 μs and are modulated by both the ratio of Eu to Yb and the proportional amount of Gd separating them. The full effect of composition on visible lifetime, as well as the correlations for estimating intermediate values, is shown in Fig. 4d. In revealing the effect of composition on lifetime, we can multiplex composition and lifetime to develop a continuum of compositions for any desired lifetime and vice versa.

The combination of metals present in each compound also has a significant effect on lifetime in the NIR. Supplementary Fig. 7a shows that diluting Yb with Gd has little effect on the lifetime of each compound, but the inclusion of Eu results in a significantly more complex decay with a long, low amplitude tail, taking tens to hundreds of times longer to fully decay (Supplementary Fig. 7b). This change in decay rates occurs because, after the initial burst of energy from the linker excitation is gone, the long-lived Eu continues to feed the short-lived Yb.

The NIR emission decay curves were accurately modeled as biexponential decays. While the biexponential decay curve is well known, the typical least squares minimization used for fitting is necessarily iterative and can be complicated when applied to equations with multiple fit parameters[45]. A method was thus developed that enables the two decay times and the amplitude to be computed from three parameters that are numerically derived from the experimental decay curves (initial inverse decay rate, average decay time, and square root of the second moment of the decay). This moments method is described in detail in Supplementary Fig. 17.

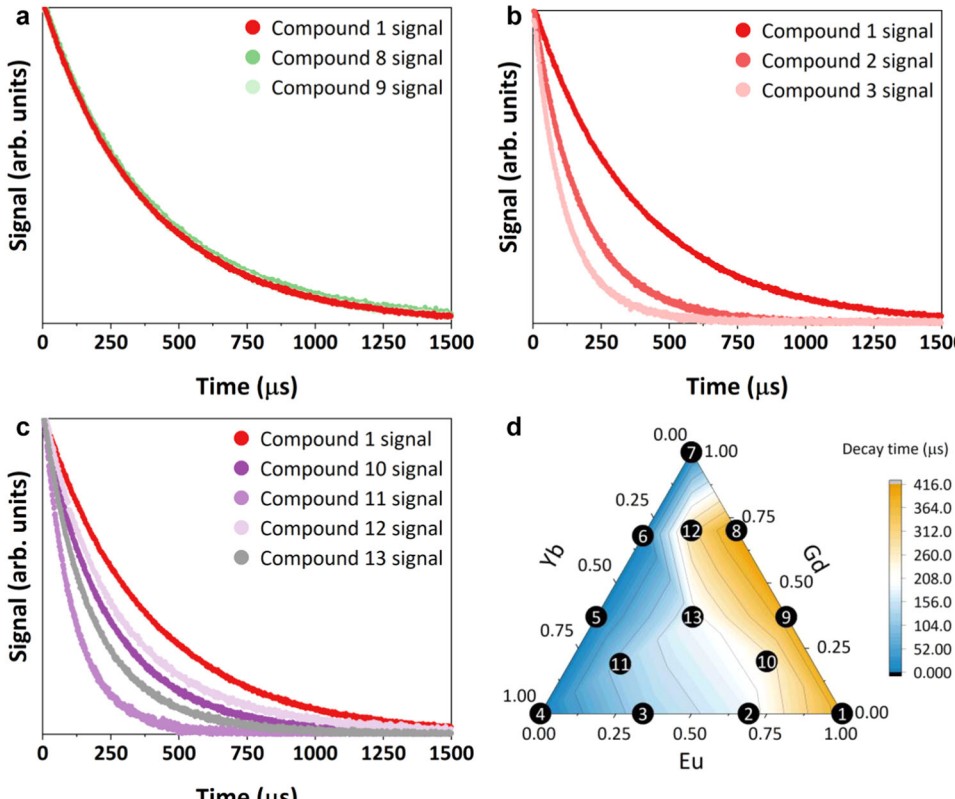

**Fig. 4 | Fluorescence decay data for reported compounds. a–c** Visible decay curves for all compounds containing Eu; **d** A ternary diagram of the different compounds with colors corresponding to the lifetime of each. These show the relative effects of Gd and Yb content on emission lifetimes of Eu in these materials.

### Table 2 | Characteristic times for the single exponential decay of each compound that emits visible light

| Compound | 1 | 2 | 3 | 8 | 9 | 10 | 11 | 12 | 13 |
|---|---|---|---|---|---|---|---|---|---|
| τ (µs) | 367 | 182 | 125 | 415 | 390 | 240 | 122 | 300 | 189 |

For these measurements, $\lambda_{ex}$ = 337 nm, and $\lambda_{em}$ = 620 nm. The standard deviation of reported lifetimes is +/−0.22% based on 5 measurements of a single compound. The uncertainty of the measurements reported is +/−5% based on repeated measurements across multiple systems and calibration against published materials.

### Heterometallic distribution discussion

Having demonstrated that changes in lifetime are a consequence of both the ratio of the Eu donor and Yb acceptor, and the presence of the optically inactive Gd ions, we note that the dependence on all three creates a broad encoding space, with multiple possible compositions for a given lifetime and vice versa.

In the dimetallic compounds, the interactions between each RE are relatively simple because the inclusion of the inactive Gd ion does not significantly affect the lifetime of either Eu or Yb, as it does not participate in energy transfer. In other studies, dilution of a visible emitter by an inactive ion has been shown to affect the luminescence intensity or quantum yield of the compound examined, but its effect on the compound's lifetime was never explored[31].

When the compound contains Eu and Yb, the lifetime of both visible and NIR emissions are affected. Because the $^2F_{5/2}$ excited state of Yb lies below the $^5D_0$ excited state of Eu, energy is transferred non-radiatively from Eu to Yb. Yb has a significantly shorter lifetime than Eu and emits the transferred energy in the NIR range. The draining of energy from the high energy Eu excited states to the low energy Yb excited states causes a significant reduction in the visible lifetime and a simultaneous lengthening in the NIR lifetime. The relative lifetimes of each compound in this case are determined only by the ratio of Eu to Yb present in each.

In trimetallic compounds, the Gd remains inactive, but provides physical spacing between the Eu and Yb ions. While the position of the Gd cannot be controlled precisely at the atomic level, its presence in the MOF will increase the average distance between the emitting ions and reduce the statistical chance of energy transfer between them. This modulation effect is most visible at the extremes of composition. For example, the equimolar compound **13** has a characteristic visible lifetime of 189 µs, but the addition of a significant amount of Eu (compound **10**) extends the visible lifetime to 240 µs, and a further addition of Gd (compound **12**) increases the visible lifetime to 300 µs. Because the physical separation caused by Gd effectively prevents energy transfer between Eu and Yb, geometric proximity has a greater impact on lifetime than the Eu-to-Yb ratio alone.

We demonstrate this concept using an averaging approach to metal distribution at the individual cluster level. Based on the evidence provided by the lifetime measurements, Fig. 5a visually represents mixed metal clusters. We determined the average cluster composition for each compound by dividing the nine metal ions according to the bulk composition and randomizing their position in the cluster to highlight a representative cluster for each composition. This visualization explains how different compositions can produce identical lifetimes in a three-metal system, and how compositions with identical quantities of visible emitters can have vastly different lifetimes. For example, compounds **11** and **12** contain nearly identical quantities of Eu, but the presence of Gd in compound **12** acts as a barrier to energy transfer and dramatically lengthens the lifetime.

To gather additional insights into how heterometallic compositions and relative atomic distribution impact the electronic structure of the materials, further investigation has been conducted using

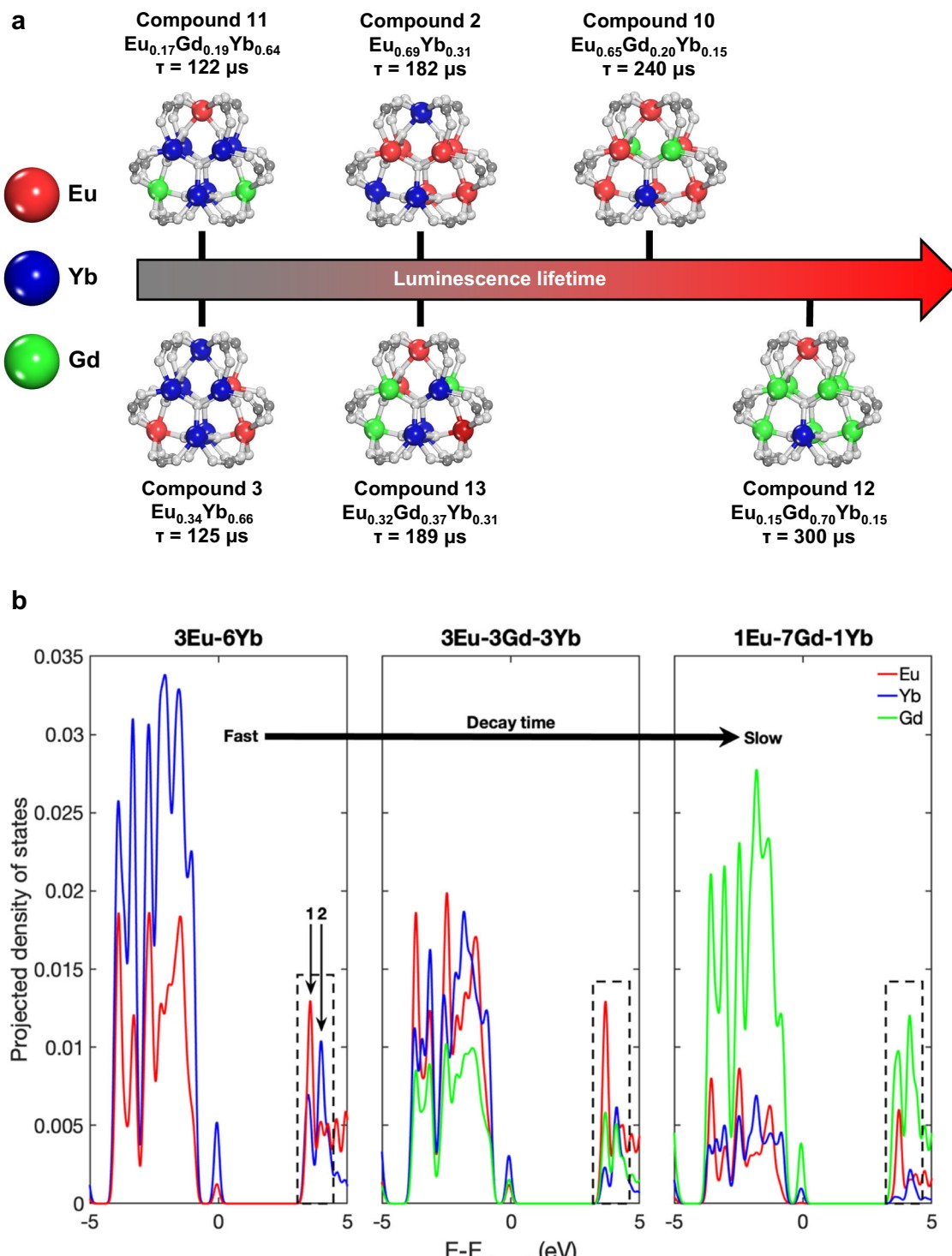

**Fig. 5 | Information regarding metal distribution and its effect on the reported MOFs. a** Illustration of the effect of composition on visible lifetime for different compounds. The lifetime of a compound is determined by the probability of a Yb metal center (blue sphere) neighboring an Eu metal center (red sphere) and receiving energy from it. The presence of a Gd metal center (green sphere) reduces the probability that an Eu center will have a neighboring Yb center. **b** Calculated RE PDOS near the electronic band edges for three representative heterometallic clusters 3Eu-6Yb (left), 3Eu-3Gd-3Yb (center), and 1Eu-7Gd-1Yb (right). The PDOS identify the relative electron density localized on each of the three RE elements; Eu (red), Yb (blue), and Gd (green). Each panel highlights the two primary peaks of RE electron density in the conduction band (black dashed box with labeled vertical arrows 1 and 2) which participate in excited state relaxation mechanisms. The three panels are organized by relative luminescence decay time as measured in this experimental work.

density functional theory (DFT) calculations. Projected density of states (PDOS) calculations are well suited to provide individual elemental contributions to the overall homometallic and heterometallic cluster electronic structures, thus helping elucidate the complex correlation between metallic compositions and resulting unique photophysical signatures. Accordingly, heterometallic clusters were modeled for the six compounds highlighted in Fig. 5a, along with the three homometallic clusters, Figs. S11 and S12, to identify qualitative

information in correlating decay times with metallic distributions. For simplicity, the compounds in Fig. 5a can be categorized by decay rate as fast (compounds **3** and **11**), intermediate (compounds **13** and **2**), and slow (compounds **12** and **10**). Models for compounds **3**, **13**, and **12** are presented as exemplars for the three luminescence lifetimes regimes to highlight the impact of heterometallic distribution, Fig. 5b.

The calculated electronic structures of all metallic clusters identify two primary peaks near the edge of the conduction band and are clearly marked in Fig. 5b. To emphasize the correlation of metallic distribution with electronic structure, in Fig. 5b are presented the RE only PDOS for clarity. Notably, the peaks contain contributions from all elements within the cluster, with large densities localized on C and O atoms, Supplementary Fig. 12. The hybridization of the organic components with the RE species is expected as the metallic clusters are bridged via organic linkers in the periodic material.

The results of the calculated electronic structures support two primary trends identified in the experimentally observed spectra: (i) increased Eu content results in longer luminescence lifetimes and (ii) increased Gd content disrupts intermetallic energy transfer between Eu and Yb. The calculated PDOS for models shows a matching trend of increasing Eu:Yb density at the first conduction band peak and models from fastest to slowest luminescence decay, Fig. 5b and Supplementary Fig. 11. The higher PDOS of Eu at the lower energy state indicates that longer lived charge carriers nonradiatively migrate to Eu prior to any optical emission event, supporting the observed experimental results.

The introduction of Gd into the heterometallic clusters is hypothesized to impact the optical response by increasing spatial separation between the optically active Eu and Yb species in the individual clusters. Spatial separation reduces the transition dipole moment between Eu and Yb species, reducing the likelihood of the event. Electronic structure calculations reinforce this premise revealing that introduction of Gd into the heterometallic gives rise to new electronic states. Importantly, the presence of Gd directly impacts the increased Eu:Yb ratio at the first conduction band peak, Fig. 5b, further supporting long lived charge carriers nonradiatively relaxing to Eu and resulting in long lived photoluminescence lifetimes.

### Encoding via luminescence lifetime

Because the time scale of luminescence decay in these materials is on the scale of tens to hundreds of microseconds in both the visible and NIR ranges, it can be decoded with relatively unsophisticated interrogation equipment, unlike tags with nanosecond-scale lifetimes[2,3]. Further, because the lifetime encoding is invisible to the unaided eye, the tag can also be covert, visible only to those who know the correct method of interrogation.

To demonstrate the relative simplicity of interrogating these highly complex tags, we used a microplate reader to measure standard and time gated (measured a specified time after a pulse of excitation) fluorescence signatures. As proof-of-concept, we utilized the braille alphabet and two representative trimetallic compositions displaying very different decay profiles: compound **11** (short lifetime) and compound **12** (long lifetime). Utilizing the braille alphabet as encoding method is advantageous because it is already a coded form and is also less complex than using a dot matrix for each letter.

To facilitate suspension of the MOF particles in ethanol, we mechanically ground the MOFs to reduce particle size. Supplementary Figs. 8 and 9 show the PXRD spectra and lifetimes of these ground samples and confirm that the size reduction did not interfere with their crystallinity or decay dynamics. Supplementary Fig. 10 shows the PL spectra of both ground samples, which are indistinguishable to the microplate reader.

In Supplementary Figs. 13 and 14 we show two simple tags, based on single compound compositions, using steady-state fluorescence excitation and emission to encode different three letter sequences.

This method encodes based on emission alone (without differentiating by lifetime) and could be interrogated using a microplate reader or simply visually examined using a UV lamp.

We then created a dynamic tag, which presents a nearly uniform fluorescence signature (an undifferentiated $3 \times 6$ block of dots) under the standard static excitation and emission method; the braille code is only revealed if measurements are taken with the correct delay after excitation (Fig. 6a). The measured intensities for each well at different time points is shown in Supplementary Fig. 15. Correct interrogation of the dynamic tag would require a threshold operation at a specified time and intensity (above which a dot is considered filled and below which a dot is considered blank). For example, Supplementary Fig. 15 shows that this threshold value could be set anywhere between 1000 and 3000 arb. units after a 750 µs delay.

As a final test, we overlaid two separate 3 letter codes, using standard fluorescence excitation and emission for one signature and time-gated fluorescence for the second signature to create a double-encoded dynamic tag. Figure 6b and Supplementary Fig. 16 show a tag utilizing two different compounds overlaid within the same set of $3 \times 6$ wells. In this case, one pattern of dots is brightest under standard excitation and emission or after short delays because of the concentration of compounds present. However, when measurements are taken after a delay, the brightness becomes dependent on lifetime rather than concentration, and a different pattern is shown. As in the dynamic tag above, both a correct threshold value and a correct time delay are required in order to decode the tag. Numerical raw data for all encodings is given in Supplementary Tables 2–4.

Materials with tunable lifetimes could be directly incorporated into printable inks to provide authentication for security documents, packaging, etc. To demonstrate this process, we patterned a transparent thunderbird logo containing discrete regions of short and long lifetime pigments (compounds **11** and **12**) onto a transparent substrate (Fig. 7a–e). Examination with a static UV light source reveals the logo in its entirety (Fig. 7e), but differentiation of the distinct materials requires a high frame-rate camera. Digital high-speed imaging of the logo after pulsed excitation, (Fig. 7g), shows the distinct decay rates of each compound, with decay profiles that match what is expected based on prior measurements of the compounds alone (Fig. 7h). A video showcasing the real-time high-speed imaging of the Sandia thunderbird logo tag is shown in the Supplementary Movie 1. This real-life demonstration supports both the effective control over the lifetime and the utility of that control in a practical setting for tag materials. These patterns were generated by laser direct write printing and the use of a laser-cut stencil, but could be readily adapted to other approaches, including flexography/(micro) stamping, ink-jet, and additive manufacturing (e.g. direct-ink write).

While not demonstrated here, additional layers of encoding could be introduced either through the presence of different emitting REs or their ratios (i.e. differentiating the encoding via spectra or color), through the presence of inactive REs like Gd that are detectable via X-ray fluorescence or EDS (i.e. differentiating by composition independent of the measurable spectra), or through the inclusion of additional molecules in the MOF pore-space. The continuum of lifetimes and compositions displayed in Fig. 4d allows composition, spectrum, and lifetime to be decoupled from each other, dramatically expanding the potential space for multilayered encoding. The scale of the visible lifetimes and presence of NIR signal provide examples of covert layers of encoding, in addition to the overt encoding based on visible emission.

In summary, here we have demonstrated the use of a ternary structural space to effectively decouple composition and photophysical properties for the creation of highly complex, difficult to counterfeit optical tags. Composition and lifetime are decoupled, ensuring true orthogonality in encoding and enabling the synthesis of a large set of rationally designed materials.

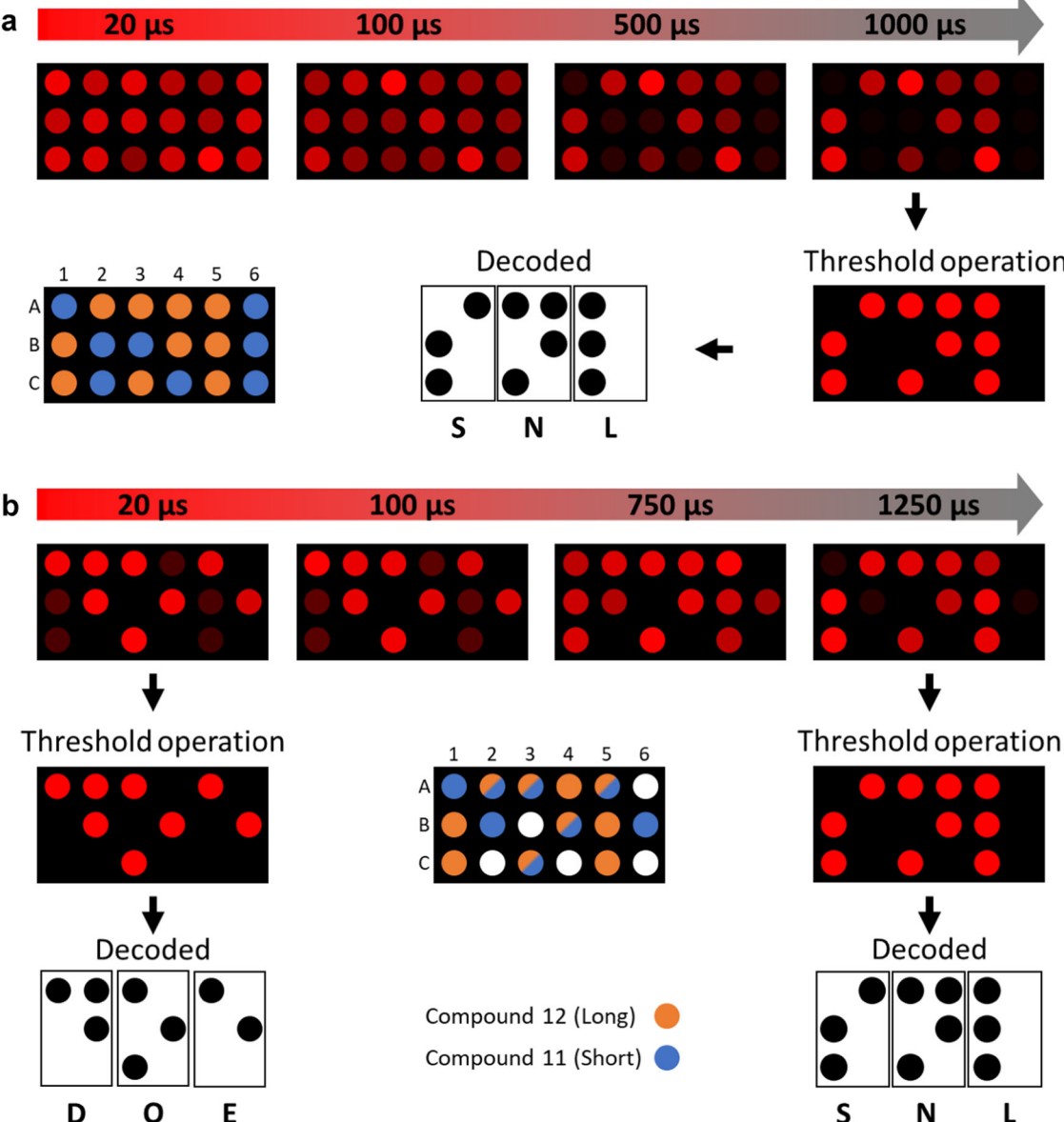

**Fig. 6 | Representation of an encoded message utilizing the braille alphabet in a 96-well plate. a** A dynamic tag that shows the transition from undifferentiated dots to the message over time based on different compound lifetimes. **b** A double-encoded dynamic tag that shows an initial encoding based on emission intensity via compound concentration in each well, with a final encoding based on different compound lifetimes. The intensity of each red dot is based on experimental data. Threshold operations are the process of choosing an intensity value as a cutoff for reading a dot as lit versus unlit for the purpose of decoding.

Precise manipulation of the luminescence decay dynamics over a wide microsecond regime was achieved owing to the control over metal distribution in these systems. The lifetime is efficiently controlled by energy transfer between visible (Eu) and NIR (Yb) emitting metals, while an optically inactive metal (Gd) dilutes them to facilitate or hinder that transfer.

To demonstrate the utility of these materials as practical optical tags we implemented two design strategies, leveraging visible lifetimes in the 100 s of µs. The first encoding strategy utilized the braille alphabet, interrogated by a standard laboratory microplate reader to produce 3 tags: (i) a static pattern read under constant illumination; (ii) a dynamic pattern that appears undifferentiated under constant illumination but is revealed via lifetime of its constituents; (iii) a double dynamic pattern that utilizes both concentration and lifetime to change the encoded message depending on when the pattern is read. The second strategy utilized photocurable inks consisting of

two distinct compositions patterned on glass and interrogated via digital high-speed imaging to produce a dynamic image that changed over time. These proof-of-concept demonstrations show the utility of these tag materials in facile, complex encoding and interrogation, both accessible for widespread use and secure against counterfeiters. The material design strategy presented can potentially be applied to a variety of rare-earth MOFs with different organic or metallic constituents, expanding the library of unique, multilayered encodings

## Methods

### Synthesis of compounds 1–13

In a typical synthesis, 0.0229 mmol of metal salts ($EuCl_3·6H_2O$, $Gd(NO_3)_3·6H_2O$, and $Yb(NO_3)_3·5H_2O$, Sigma-Aldrich, 99.99% trace metal basis) in the same proportions as the target composition, 0.0065 mmol (3.65 mg) of TCPB (Sigma Aldrich, contains up to 6 wt%

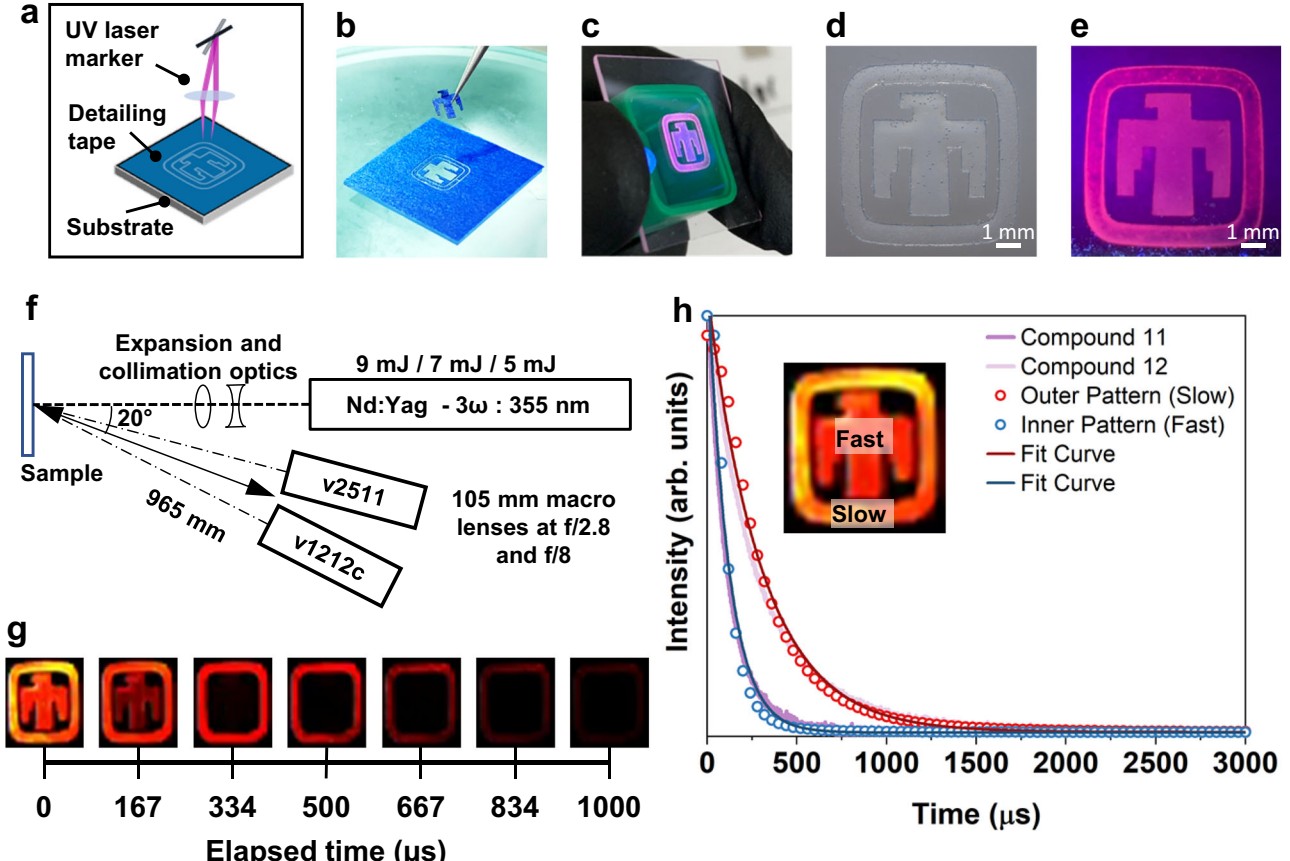

**Fig. 7 | A demonstration of patterning and interrogating a tag made with compounds 11 and 12 in an ink. a** Diagram detailing stencil design and etching. **b** Stencil composed of painters' tape on glass. **c** Inked Sandia thunderbird logo under UV light, held in hand for scale. **d**, **e** Inked logo under ambient and UV light respectively. **f** Diagram of the laser excitation and digital high-speed imaging setup used to capture the decay of the logo. **g** Time-lapse of tag emission after pulsed laser excitation, showing two distinct lifetimes for the thunderbird and border respectively. **h** Decay data curves and fit curves for each section of the logo, derived from digital high-speed images, compared to the decay of pure compound **11** and **12** reported above. In every image, the pattern was printed on a 25 mm × 25 mm substrate.

water, ≥98%), 4.0075 mmol (561.49 mg) of 2-fluorobenzoic acid, (Sigma Aldrich, 97%), 0.2 mL of $HNO_3$ (3.5 M in DMF, Fisher Chemical, Certified ACS plus), and 2.8 mL of dimethylformamide (DMF, Acros Organics, 99.5% for HPLC) were combined in a glass vial. This mixture was heated to 115 °C over one hour, held there for 48 h, then cooled to room temperature. The resulting particles were washed three times with DMF, then exchanged with methanol over three days (the methanol was changed daily).

### X-ray single-crystal diffraction data collection and determination

The X-ray diffraction data were measured using a Bruker D8 Venture dual-source single crystal X-ray diffractometer, employing Cu Kα radiation ($\lambda$ = 1.54178 Å). Data were collected using a Photon II CMOS area detector on samples superglued to glass fibers. Reflection data were processed using the Bruker Apex 3 Suite, including an absorption correction through SADABS, and space group determination using XPREP. The structure was determined with SHELX-2019 software. The structure was solved with XT using intrinsic phasing and refined using full-matrix least-squares on F2 through XL. The structure was finalized in Olex 2 (v.1.3.0).

### Thermogravimetric analyses (TGA)

Measurements were conducted on a SDTQ600 TA instrument. The samples were heated to 900 °C at a 5 °C min⁻¹ heating rate, under continuous nitrogen flow.

### Powder X-ray diffraction

Measurements were performed on a Bruker D2Phaser instrument using CuKα radiation ($\lambda$ = 1.54178 Å).

### Scanning electron microscopy and energy dispersive spectroscopy

SEM images were captured using an FEI NovaNano SEM 230 at an accelerating voltage of 30 kV. EDS analyses were collected using an EDAX Genesis Apex 2 with an Apollo SDD detector.

### Photoluminescence measurements

The visible PL emission spectra of powder samples were collected using a Horiba Jobin-Yvon Fluorolog-3 double-grating/double-grating Fluorescence Spectrophotometer in front-face mode. This instrument uses a continuous wave 450 W Xenon arc lamp excitation source and a room temperature R928P photomultiplier tube (PMT) for the detection of visible light. A complete instrumental correction was performed on all spectra which compensates for factors such as the grating efficiencies and the wavelength-dependent PMT response. The powder samples were sandwiched between glass slides for these measurements.

### Photoluminescence lifetime measurements

PL lifetime measurements were made on powder samples sandwiched between quartz slides by exciting the samples with a PTI GL3300 pulsed nitrogen laser at 337 nm. Two detectors were used, one to trigger the

oscilloscope (Thorlabs PDA55 silicon photodetector) and the other to measure the emission. The signals were averaged using a 1 GHz Tektronix TDS 5104 digital oscilloscope. A Thorlabs PDA55 silicon photodetector was used to measure the Eu$^{3+}$ emission. A 620 nm narrow bandpass filter with FWHM of 10 nm was placed in front of the detector. A Thorlabs PDA10CS InGaAs switchable gain amplified detector was used to measure the Yb$^{3+}$ emission. A lens and two long-pass filters (550 nm and 950 nm cut-on wavelength) were placed in front of the detector. The uncertainty of the measurements reported is +/−5%.

## Absolute quantum yield (QY) measurements

QY was determined by exciting the sample with diffuse light inside an integrating sphere[46]. The powder samples were placed in Pyrex NMR tubes for these measurements and inserted into the integrating sphere. Both the diffuse excitation and emission light power were simultaneously recorded. The QYs were calculated from these power measurements.

## Microplate reader fluorescence and time-gated fluorescence measurements

To facilitate equal distribution of sample into wells, compounds **11** and **12** were activated at 120 °C for 12 h, then ground by hand using a mortar and pestle to reduce particle size. Both were then resuspended at known concentrations in ethanol. Ground compounds **11** and **12** were placed in a 96-well black plate for analysis. Fluorescence and time-gated fluorescence measurements were made using a Biotek Neo 2 microplate reader with excitation of 340 nm (+/−20 nm) and emission set of 617 nm (+/−20 nm). Standard fluorescence measurements were made with simultaneous excitation and emission. Time-gated fluorescence measurements were made with the excitation separated from the emission collection. Emission was collected over 20 μs and collection was initiated at 20, 100, 750, 1000, or 1250 μs after excitation.

Ground versions of compounds **11** and **12** were first examined independently using standard fluorescence measurements. For these experiments, 1 mg of sample was placed into each well, and samples were arranged using the braille alphabet to encode DOE or SNL with the raised dots represented by filled wells. Next, tags with separate applications of ground compounds **11** and **12** were then tested to create a hidden signature of SNL in the braille alphabet, encoded with time-gated fluorescence. To test this, the wells encoding for the raised dots were filled with 0.2 mg of ground compound **12** and the empty spaces were filled with 1 mg of ground compound **11**. Volumes of ethanol were kept even between the wells, and the plate was read using the time-gated fluorescence settings. Finally, overlaying the DOE and SNL signatures were tested. For this experiment 1 mg of compound **11** was laid out as DOE and then overlaid with 0.1 mg of compound **12** laid out as SNL. The plate was read using the time-gated fluorescence settings.

## Materials patterning

For the thunderbird pattern, ground compounds **11** and **12** were mixed at an 8–10% weight ratio with a UV-curable optical adhesive (NOA 61 or 65; Norland Products) and deposited sequentially on a fused silica coverglass on desired areas of an exposed stenciled pattern. The stenciled image was laser patterned onto a film (blue painter's tape; 3 M) adhered to the substrate using a UV laser marker (Keyence MD-U). Material deposited on the stencil was cured with a UV spot lamp (Dymax RediCure).

## High-speed imaging experiments

The light source used was a Quantel Q-smart Twin laser set, run independently to produce a single pulse. The beam energy was 5 mJ per pulse, at a repetition rate of 10 Hz. The initial beam diameter of 9 mm was expanded using a plano-concave lens (Thorlabs LC4252, $f = −30$ mm) and an achromatic doublet lens (Thorlabs AC508-150,

$f = 150$ mm). An iris was placed after the lenses to generate a 23 mm diameter beam that was incident upon the sample.

For the lifetime measurement, two high-speed cameras were used: a broadband (no Bayer color filters in front of the pixels) Vision Research v2511 running at 25,000 frames per second (fps), with an exposure of 39.54 μs; and a color Vision Research v1212c running at 12,000 fps with an exposure of 82.59 μs. Both cameras used Nikkor 105 mm macro lenses at f/2.8 for the small tag and f/8 for the large one. The cameras resolution was set at their maximum of 1280 × 800 pixels.

## Decay curve fitting

Most of the luminescent decay curves collected for this paper were accurately modeled as double exponential decays. Double exponential decays have five parameters, two amplitudes, two decay times and a baseline amplitude. The baseline was obtained from the detector from the signal just before the laser pulse. The laser pulse rate was about 5 Hz, so this corresponds to a signal delay of ~200 ms after the pulse, which is many times longer than the longest decay time measured. The two amplitudes were reduced to one by normalizing the luminescent decay curve to unity. The decay model can then be written

$$f(t) = ae^{-t/\tau_1} + (1-a)e^{-t/\tau_2}. \tag{1}$$

This equation can be fit to the experimental decay curve by least squares minimization, but this approach has well-known difficulties, notably local minima and non-positivity of the three parameters. We have thus developed an approach for the extraction of the three parameters that does not involve least squares fitting.

We begin by defining three parameters that are obtained numerically from the experimental decay curve. These parameters will then be used to compute the decay parameters, i.e., the two decay times and the amplitude $a$. The first parameter $A$ is the inverse initial decay rate.

$$A^{-1} \equiv \lim_{t \to 0} \frac{-d \ln f(t)}{dt} = \frac{a}{\tau_1} + \frac{(1-a)}{\tau_2} \tag{2}$$

The second parameter is average decay time $B$.

$$B = \lim_{t \to \infty}\left[B(t) \equiv \int_0^t f(s)ds\right] = a\tau_1 + (1-a)\tau_2 \tag{3}$$

Finally, the parameter $C$ is the square root of the second moment of the decay.

$$C^2 = \lim_{t \to \infty}\left[C^2(t) \equiv \int_0^t sf(s)ds\right] = a\tau_1^2 + (1-a)\tau_2^2 \tag{4}$$

$A$, $B$, and $C$ are defined so that they each have the units of time and it is easily shown that $A < B < C$. To obtain accurate integrals one must know the baseline accurately and one should ideally extrapolate to infinite time. This extrapolation can easily be done by plotting $B(t)$ and $C(t)$ against the amplitude of the decay. But if data is collected over a long enough time, then it is sufficient to make a parametric plot of $B(t)$ against $C(t)$ and obtain $B, C$ from the endpoint. Plotting in this way greatly reduces noise.

The solution to these three equations is a quartic, but there is a simpler way to solve these. The second decay time can be expressed in terms of the first by $\tau_2 = \frac{(B - a\tau_1)}{(1-a)}$ and by $\tau_2^2 = \frac{(C^2 - a\tau_1^2)}{(1-a)}$. Combining these gives a quadratic equation whose solutions are

$$\tau_1 = B \pm \sqrt{B^2 + \frac{B^2 + (1-a)C^2}{a}} \tag{5}$$

Any given value of the amplitude will then generate a value of $\tau_1$, which can then be used to compute a value of $\tau_2$. These are the correct values if they satisfy Eq. 2. A plot of $g = A - 1/[\frac{a}{\tau_1} + \frac{(1-a)}{\tau_2}]$ vs $a$ must therefore exhibit a zero crossing, which gives the correct value of the amplitude, and thus the two decay times. To remove negative values and singularities it is best to plot the fit quality function $Q = \text{atan}(g^2)$. Plotting this logarithmically creates a line that points directly to the correct amplitude.

At this point an example of extracting the decay parameters from the experimentally derived parameters is instructive. Let the experimental parameters be $a = \frac{1}{3}$, $\tau_1 = 1$, $\tau_2 = 6$. Then $A = \frac{9}{4}$, $B = \frac{13}{3}$, $C = \sqrt{\frac{73}{3}}$.

## Computational models and methodology

The geometry optimization and ground state electronic structures of the metallic clusters were calculated using spin-restricted DFT as implemented in the Vienna ab initio simlation package (VASP)[47,48]. The simulated cluster models were built with the following molecular formula: $[M_9(HCO_2)_{12}(\mu_3OH)_{12}O_2]^-$ $Na^+$ where M = Eu, Yb, Gd and a non-interactive Na atom is included in the simulation cell to balance the total charge of the simulation cell.

The calculations were completed in a plane wave basis set[49,50] with a 500 eV cutoff energy and converged to a force accuracy of 0.01 eV Å$^{-1}$ at the gamma point. The generalized gradient approximation (GGA) exchange-correlation functional of Perdew, Burke, and Ernzerhof designe for solids and surfaces (PBEsol)[51] was used due to previous calculation sets[52–56]. The DFT-D3 method of Grimme et al.[57]. With Becke-Johnson damping[58] was used for dispersion corrections. Projector-augmented wave (PAW) potentials[59,60] were used for C, O, H, and Na, with the large core potentials (LCPs) which represent the M(III) oxidation state used for Eu, Yb, and Gd. These potentials place the 4f electrons in the core of the potential. This approximation was utilized as an initial set of calculations to investigate the changes in core electronic structure when complex combinations of rare-earth metals are combined. The calculation set provides information on electronic interactions prior to charge being transfered to the 4f orbitals of the Eu, Yb, and Gd.

The PDOS were calculated according to the VASP protocal using LORBIT = 11. This provides spatially resolved electron density as a function of spherical volume at each atomic location. The PDOS calculations utilized the standard RWIGS values provided in each elements PBE POTCAR file. While spherical limits on electron density are known to underpredict hybridized electrons in space, the primary atomic contributions calculated in the PDOS were confirmed by vizualizing molecular orbitals from calculated PARCHG files which are derived from the calculated wavefunction.

*Statistics and reproducibility:* The standard deviation of reported lifetimes is +/−0.22% based on 5 measurements of a single compound. The uncertainty of the measurements reported is +/−5% based on repeated measurements across multiple systems and calibration against published materials.

### Reporting summary

Further information on research design is available in the Nature Portfolio Reporting Summary linked to this article.

## Data availability

Crystallographic data for the structure reported in this Article have been deposited at the Cambridge Crystallographic Data Centre, under deposition numbers CCDC 2225329. Copies of the data can be obtained free of charge via https://www.ccdc.cam.ac.uk/structures/. The detailed data for the study is available from the corresponding author upon request.

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

## Acknowledgements

The authors would like to thank Dr. Eric Sikma, Luke Lucero, Dr. Nichole Valdez, and Dr. Mark Rodriguez for support with materials synthesis and characterization; Dr. Caroline Winters for her assistance with the high-

speed photography experiments; and Dr. Jacob Harvey for support with structure optimization for computational studies. This research was supported by the Laboratory Directed Research and Development Program at Sandia National Laboratories (D.S.G.) and was performed, in part, at the Center for Integrated Nanotechnologies, an Office of Science User Facility operated for the U.S. Department of Energy (DOE) Office of Science. This article has been authored by an employee of National Technology & Engineering Solutions of Sandia, LLC under Contract No. DE-NA0003525 with the U.S. Department of Energy (DOE). The employee owns all right, title and interest in and to the article and is solely responsible for its contents. The United States Government retains and the publisher, by accepting the article for publication, acknowledges that the United States Government retains a non-exclusive, paid-up, irrevocable, world-wide license to publish or reproduce the published form of this article or allow others to do so, for United States Government purposes. The DOE will provide public access to these results of federally sponsored research in accordance with the DOE Public Access Plan https://www.energy.gov/downloads/doe-public-access-plan. The views expressed in this article do not necessarily represent the views of the U.S. Department of Energy or the United States Government.

## Author contributions

D.S.G. and J.I.D. developed the concept and designed the experiments. J.I.D. performed the materials synthesis and characterization and lead the manuscript writing. R.A.R. aided the materials synthesis. L.E.S., K.S.B., T.S.L., A.A.C.C., and J.E.M. supported all aspects of optical characterization. B.K. developed the materials patterning. D.J.V. carried out the DFT calculations. All the authors discussed the results and J.I.D., L.E.S., K.S.B., D.J.V., and D.S.G. wrote the manuscript.

## Competing interests

The authors declare no competing interests.
