## [Peer Review File · Nature Communications]

Reviewer comments, first round –

Reviewer #1 (Remarks to the Author):

The authors show that the compound under study shows microsecond lifetime and can be applied to lifetime-encoded tags. The mentioned 13 kinds of MOFs exhibited different lifetime thus longest and shortest one were picked up respectively to apply for encoding via luminescence lifetime. The concept is interesting, but the presentation and the explanation of the observations are not of the high standards one should expect for this journal. Also, the potential impact of the work is not convincingly presented. Therefore, the paper is not suitable to be published in Nat. Comm..

1. The introduction is too long. Nearly 1200 words make up about 20% of the paper which is unnecessary.
2. Appropriate graphic illustration for reading easily maybe should be added in introduction part.
3. Source of chemicals should be provided in supporting information. In addition, more details should be provided for proving successful synthesis of compounds.
4. In page 10, what is the exact meaning of "0,1,2,3,4" in the sentence of "when Eu is excited directly with 394 nm light, characteristic Eu emission peaks in the visible range are observed (5D₀→4F_j where j is equal to 0 at 582 nm, 1 at 592-600 nm, 2 at 617 nm, 3 at 655 nm, and 4 at 704 nm)". The represent is ambiguous.
5. Table 2 summarized characteristic times for the single exponential decay of each compound that emits visible light, the wavelength range of visible light and excitation source should be given to compare.
6. In this sentence, "In Figure S8 and S9 we show two simple tags, based on single compound compositions, using standard fluorescence excitation and emission to encode different three letter sequences." in page 12, the specific meaning of standard should be further clarified.
7. In figure 5b, the last time node is 1500 μs. However, raw data read from the double wellplate encoding at 1500 μs is missing. Similar situation also appears in figure S9. According to the supporting information, raw data read from the double wellplate encoding at 1250 μs is the last data given by the authors.
8. The current application of transparent thunderbird logo containing discrete regions of short and long lifetime didn't exhibit the outstanding properties of MOFs comparing with organic TADF materials. Further optimization is suggested to highlight the advantages of MOF porous materials.
9. The evidence of the inherent mechanism in manipulation of the luminescence decay dynamics by controlling over metal ordering in these systems is not sufficient.
10. More evidence should be supplied for photoluminescence to prove fluorescence or delay fluorescence of MOFs. Current data is insufficient.

Reviewer #2 (Remarks to the Author):

General comment:

The manuscript by Deneff, Rohwer, Butler, Kaehr, Luk, Reyes, Cruz-Cabrera, Martin, Sava Gallis present the synthesis, structural and optical characterizations of REMOF or LnTCPB compounds. This work shows a fundamental study of the luminescence mechanisms from emission spectra but especially from lifetime measurements. This well conducted study is then followed by a direct application with an encoding methodology via luminescence lifetimes. The development of these compounds from an application point of view (brail alphabet and inks for anti-counterfeiting marking) make this work a new and interesting piece of work for the scientific community. So this study is in adequacy with the "Nature Communications" journal. I recommend publication but after major revisions; improvements could be anyway realized before publication.

Comments and Minor points:

1- One point of improvement comes from the definition of compounds and their formulations. It would be useful to give the chemical formulation Ln_xTCPBy.wH₂O of the compounds as well as to give in SI perhaps the chemical representation of the ligand (chemdraw, ...). In the same way, it is not clear that in the paper that these compounds are new or already known

in the literature, the authors should clarify the situation by citing the papers that present the structures from these compounds and present a reminder of the crystal structure data of this family of compounds by giving the CIF number(s) of the structures. The calculated powder diagram used in the XRD figure 1 come inevitably from an extraction of the CIF structure file. Only a few Ln-Ln distances are provided in part 2.2 "intermetallic Energy Transfer", a quick description of the crystal structure (space group, lattice parameters, Ln environment, Ln-O distance, geometry, ligand Ln coordination), should be given in order to correlate the optical properties to the structural properties.

2- The lifetime measurements have been measured several times for each compound? The values could be indicated with uncertainty errors in the tables. The authors should at least specify whether the repeatability of the measurement has been verified.

3- No quantum yield measurements have been carried out to quantify the efficiency of the compounds with each other? The authors could justify the choice not to use this quantitative method to study luminescence mechanisms that occur in these compounds.

4- Is the thermal stability of these compounds known? The authors should recall this property if it has been investigated in previous studies or do an ATG-ATD or DSC analysis to obtain this information. For EDS analysis, the uncertainties of the measurements should be given in Table 1 (repeatability of the measures)

5- The emission spectra in Figure 2 for europium and ytterbium were recorded at 395 nm (f-f band of Eu(III)), why not record these spectra under ligand excitation at 337 nm (antenna excitation). Moreover, in this perspective, it would be nice to add the excitation spectrum or the absorption spectrum of at least one compound justifying the 337 nm value for the TCPB absorption. Moreover, the authors give 26500 cm⁻¹ as the value of the TCPB ligand, the phosphorescence spectrum allowing this value to be determined should be provided in SI. In the figure 2.b, the value of the singlet is not given while that of the triplet is given at 25200 cm⁻¹. The authors should clarify this point. What does the value at 26500 cm⁻¹ correspond to? to the singlet, to the triplet? other assignment?
Furthermore, the Eu emission spectra show the most intense band at 617 nm, so why measure the lifetimes at 620 nm? Is this an instrumental choice related to the measurement?
In the case of lifetimes, the presence of mathematical fits would allow a better understanding of the agreement between the experimental points and the fit model.

6- In the method section, it would be good to give more details on the photoluminescence measurements performed on the fluorolog-3. What sources are used, what are the detectors of the device? ...

Reviewer #3 (Remarks to the Author):

In the MS, the authors demonstrate a design strategy towards multiplexed, lifetime-encoded tags via engineering intermetallic energy transfer in heterometallic metal-organic frameworks. Compare to anticounterfeiting systems based on energy transfer between ligand and metal or fluorescence behavior of a single component, using excited lifetime for anticounterfeiting seems more complex, especially based on intermetallic energy transfer and heavily depend on the control over metal ordering. However, this referee thinks the design of this work is interesting and refreshing. I'd like to recommend the authors to make some revisions before publication.

1. Some of the introduction parts are overstated, for example, not all amorphous physical structures are difficult to characterize, some pure organic compound with clear structure have been used for anticounterfeiting.
2. Monitoring fluorescence of visible and near infrared regions simultaneously seems have higher requirements for equipment. And monitoring luminescence lifetime difference in microsecond seem difficult to popularize in practical application. The author should explain the advantage of this method.

3. Some closely related reference should be added: Nat. Commun. 2021, 12, 1363; Angew. Chem. Int. Ed. 2019, 58, 18025.

Reviewer #4 (Remarks to the Author):

In this work, the authors developed multiplexed, lifetime-encoded tags via engineering intermetallic energy transfer in heterometallic metal-organic frameworks. Manipulation of the luminescence decay dynamics over a wide microsecond regime was conducted by controlling the metal ordering in these systems. They then applied them as tags by incorporation into photocurable inks patterned on glass and interrogated via digital high-speed imaging. This work shows some important results to the field. Thus, it is recommended for publication after addressing the following issues.

1. It is better to give the chemical structures and formula of the heterometallic MOFs in the Supplementary Information for better clarity.
2. In addition to controlling the metal ordering in MOFs, is it possible to tune the organic ligands to achieve the targets as well?
3. There are basically single color tags. It will be better to achieve color-tunable tags for more advanced usage.
4. Computational studies should be performed to support the energy transfer within the selected systems.

Responses to Technical Comments:

Reviewer 1:

The authors show that the compound under study shows microsecond lifetime and can be applied to lifetime-encoded tags. The mentioned 13 kinds of MOFs exhibited different lifetime thus longest and shortest one were picked up respectively to apply for encoding via luminescence lifetime. The concept is interesting, but the presentation and the explanation of the observations are not of the high standards one should expect for this journal. Also, the potential impact of the work is not convincingly presented. Therefore, the paper is not suitable to published in Nat. Comm.

Response:

We thank the reviewer for their time and input on this work. We have addressed the concerns expressed as outlined below. We appreciate the reviewer for their time, effort, and critical read of this work. We have carefully addressed all the concerns expressed as detailed below.

Reviewer 1:

The introduction is too long. Nearly 1200 words make up about 20% of the paper which is unnecessary.

Response:

We thank the reviewer for this observation; according to this suggestion, we have streamlined the content in the introduction.

Reviewer 1:

Appropriate graphic illustration for reading easily maybe should be added in introduction part.

Response:

We appreciate the reviewer's suggestion. In the revised version of the manuscript, we have included a schematic depiction of the proposed conceptual approach in this work, Figure 1.

Figure 1. Graphical illustration exemplar highlighting the effect of controllable energy transfer on modulating the resulting lifetime in EuGdYb-based trimetallic compositions.

Reviewer 1:

Source of chemicals should be provided in supporting information. In addition, more details should be provided for proving successful synthesis of compounds.

Response:

We are happy to have the opportunity to clarify. In the methods section, “commercial sources” has been replaced with “Sigma Aldrich” to specify the vendor.

Reviewer 1:

In page 10, what is the exact meaning of “0,1,2,3,4” in the sentence of “when Eu is excited directly with 394 nm light, characteristic Eu emission peaks in the visible range are observed ($^5D_0 \rightarrow ^4F_j$ where j is equal to 0 at 582 nm, 1 at 592-600 nm, 2 at 617 nm, 3 at 655 nm, and 4 at 704 nm)”. The represent is ambiguous.

Response:

We thank the reviewer for their comment. This representation was meant to be shorthand where the “j” is replaced with the stated number to correspond to the peak at the given wavelengths. To improve the clarity of this section, in the revised version of the manuscript, the quoted section in parentheses has been replaced with a full description: “($^5D_0 \rightarrow ^4F_0$ at 582 nm, $^5D_0 \rightarrow ^4F_1$ at 592-600 nm, $^5D_0 \rightarrow ^4F_2$ at 617 nm, $^5D_0 \rightarrow ^4F_3$ at 655 nm, and $^5D_0 \rightarrow ^4F_4$ at 704 nm).” Additionally, we have made the same change in Figure 2c, labeling using the full transition notation rather than an abbreviation.

Reviewer 1:

Table 2 summarized characteristic times for the single exponential decay of each compound that emits visible light, the wavelength range of visible light and excitation source should be given to compare.

Response:

Thank you for this suggested edit. In the revised version of the manuscript, we have added the following sentence to the caption of Table 2: “For these measurements, $\lambda_{ex} = 337 \text{ nm}$, and $\lambda_{em} = 620 \text{ nm}$.”

Reviewer 1:

In this sentence, “In Figure S8 and S9 we show two simple tags, based on single compound compositions, using standard fluorescence excitation and emission to encode different three letter sequences.” in page 12, the specific meaning of standard should be further clarified.

Response:

We are grateful for the point of clarity. To better communicate the measurements taken, the word “standard” has been replaced by “*steady-state*” to reflect that the measurements were taken using constant illumination by the exciting wavelength rather than a pulse of that illumination.

Reviewer 1:

In figure 5b, the last time node is 1500 μs . However, raw data read from the double wellplate encoding at 1500 μs is missing. Similar situation also appears in figure S9. According to the supporting information, raw data read from the double wellplate encoding at 1250 μs is the last data given by the authors.

Response:

We are grateful to the reviewer for pointing out this oversight. The figure has been amended so that the last time node is 1250 μ s to reflect the raw data. To be noted, the figure numbering has been changed due to the introduction of a new figure, Figure 1.

Figure 6. Representation of an encoded message utilizing the braille alphabet in a 96 well plate. a. A dynamic tag that shows the transition from undifferentiated dots to the message over time based on different compound lifetimes. b. A double-encoded dynamic tag that shows an initial encoding based on emission intensity via compound concentration in each well, with a final encoding based on different compound lifetimes. The intensity of each red dot is based on experimental data.

Reviewer 1:

The current application of transparent thunderbird logo containing discrete regions of short and long lifetime didn't exhibit the outstanding properties of MOFs comparing with organic TADF materials. Further optimization is suggested to highlight the advantages of MOF porous materials.

Response:

We agree with the reviewer that the example presented using the thunderbird is relatively simple. We felt that this was appropriate given the focus of the paper on manipulating lifetime specifically and intended to show that measurement using high speed photography is possible at the time scales and quantum yields involved.

The text after Figure 6 includes a statement regarding the extensibility of MOFs as a system, and has been amended to additionally address the porosity of MOFs (changes highlighted): *“While not demonstrated here, additional layers of encoding could be introduced either through the presence of different emitting REs or their ratios (i.e. differentiating the encoding via spectra or color), through the presence of inactive REs like Gd that are detectable via X-ray fluorescence or EDS (i.e. differentiating by composition independent of the measurable spectra), or through the inclusion of additional molecules in the MOF pore-space.”*

With respect to comparisons with organic TADF materials, we thank the reviewer for bringing attention to this interesting subject. We feel that a direct comparison of our materials with TADF materials would be difficult because of the inherently different mechanisms involved, intersystem crossing for TADF materials and ligand to metal or metal to metal charge transfer for MOFs. We do not examine ligand lifetime in this work, and are only concerned with it as an antenna for the metal centers.

However, to acknowledge this relevant research topic in the broader context of other types of materials with tunable photoluminescent response, we have added a sentence to the introduction when discussing lifetime: *“...additional data. For example, organic compounds with variable lifetime have been reported, including thermally activated delayed fluorescence materials 2,27 However...”* and have also added a relevant citation for this class of materials.

27. Yang, Z.; Mao, Z.; Xie, Z.; Zhang, Y.; Liu, S.; Zhao, J.; Xu, J.; Chi, Z.; Aldred, M. P., Recent advances in organic thermally activated delayed fluorescence materials. Chemical Society Reviews 2017, 46 (3), 915-1016.)

Reviewer 1:

The evidence of the inherent mechanism in manipulation of the luminescence decay dynamics by controlling over metal ordering in these systems is not sufficient.

Response:

We thank the reviewers for the point of clarification. We believe confusion has arisen out of the use of the term “ordering” specifically. To clarify and improve the communication of the intended mechanism of control, we have changed occurrences of the term “ordering” to “distribution” with respect to the metal. We believe this more accurately reflects our intent, and avoids the confusion expressed by the reviewers. Further, we have included extensive electronic DFT calculations to support the luminescence mechanisms proposed in this work.

Reviewer 1:

More evidence should be supplied for photoluminescence to prove fluorescence or delay fluorescence of MOFs. Current data is insufficient.

Response:

We thank the reviewer for their comment and for the opportunity to clarify the photoluminescence mechanisms detailed in our manuscript. As in our response to comment 8, we believe that there is a difference in the mechanism in question given our work's focus on the metal emissions specifically. For example, TADF or other organic based materials display prompt and delayed fluorescence depending on whether primary singlet emission, triplet emission, or reverse-intersystem crossing singlet emission is being measured. In the case of the MOFs reported here, only the ligand emission could be characterized in this way, and we do not report the lifetime of the ligand emission. The lifetime of each material is based solely on the emission or energy transfer characteristics of the metals involved.

To clarify our meaning, however, we have replaced the term “delayed fluorescence” with “time gated fluorescence” to reflect the fact that we are taking measurements with a time gap between excitation and measurement, rather than implying that the fluorescence itself is delayed. The decay is continuous for our measurements as only a single pulse of excitation is used, but the measurement is delayed by different time intervals after excitation to capture the effect of the decay on the appearance of the material.

Reviewer 2:

The manuscript by Deneff, Rohwer, Butler, Kaehr, Luk, Reyes, Cruz-Cabrera, Martin, Sava Gallis present the synthesis, structural and optical characterizations of REMOF or LnTCPB compounds. This work shows a fundamental study of the luminescence mechanisms from emission spectra but especially from lifetime measurements. This well conducted study is then followed by a direct application with an encoding methodology via luminescence lifetimes. The development of these compounds from an application point of view (brail alphabet and inks for anti-counterfeiting marking) make this work a new and interesting piece of work for the scientific community. So this study is in adequacy with the “Nature Communications” journal. I recommend publication but after major revisions; improvements could be anyway realized before publication.

Response:

We appreciate the reviewer's positive outlook on this work and have addressed in detail all the concerned raised below.

Reviewer 2:

One point of improvement comes from the definition of compounds and their formulations. It would be useful to give the chemical formulation $\text{Ln}_x\text{TCPBy}\cdot n\text{H}_2\text{O}$ of the compounds as well as to give in SI perhaps the chemical representation of the ligand (chemdraw, ...). In the same way, it is not clear that in the paper that these compounds are new or already known in the literature, the authors should clarify the situation by citing the papers that present the structures from these compounds and present a reminder of the crystal structure data of this family of compounds by giving the CIF number(s) of the structures. The calculated powder diagram used in the XRD figure 1 come inevitably from an extraction of the CIF structure file.

Only a few Ln-Ln distances are provided in part 2.2 "intermetallic Energy Transfer", a quick description of the crystal structure (space group, lattice parameters, Ln environment, Ln-O distance, geometry, ligand Ln coordination), should be given in order to correlate the optical properties to the structural properties.

Response:

We appreciate the reviewer's suggestion to provide additional information on the structural and chemical formulations for these compounds. To clarify, the representative crystal structure for Y and YbNd-based compounds was previously detailed elsewhere by us and others, (references 17 and 33), while the crystal structure for the Eu analog is reported for the first time with this work. We have included a paragraph to capture this aspect more accurately:

“The representative crystal structure for these REMOFs was previously detailed elsewhere by us and others;^{17,18} the crystal structure for the Eu analog is reported for the first time herein. Briefly, the three periodic structure is derived from nonanuclear clusters linked by 12 carboxylate groups of the TCPB linker. The resulting framework possesses intrinsic porosity accessible via 1D channels of ~ 1.2 nm.”

Furthermore, we have included a schematic that highlights the representation of the linker (Figure 1). As additional supporting information for this work, we have included the cif file for Eu-based compound **1**, reported in this manuscript for the first time. The crystallographic information has been included in Table S1, along with a general formula:

*In monometallic compositions, M= Eu, Yb, or Gd (compounds **1**, **4**, **7**). For dimetallic and trimetallic compositions (compounds **2**, **3**, **5**, **6**, **8**, **9**, **10**, **11**, **12**, **13**) the metal distribution is a fractional combination of those three metals based on each individual composition, as outlined in **Table 1**.*

Table S1. Crystal data and structure refinement for compound **1**.

Empirical formula	C204 Eu18 O93.26
Formula weight	6677.48
Temperature	116 K
Wavelength	1.54178 Å
Crystal system, space group	Hexagonal P 6 ₃ /m m c
Unit cell dimensions	a = 22.0812(3) Å b = 22.0812(3) Å c = 25.2784(7) Å
Volume	10674.0(4) Å ³

Z, Calculated density	1, 1.039 g/cm ³
F(000)	3104.0
Crystal size	0.34 x 0.18 x 0.12 mm
Theta range for data collection	2.310 to 55.970°
Reflections collected / unique	2600/2377
R indices	R1 = 0.1353, wR2 = 0.3167
Largest diff. peak and hole	-4.752 to 6.088 eÅ ⁻³

Finally, the methods section has been updated with details on the X-ray single-crystal data collection and determination.

X-ray single-crystal data collection and determination. *The X-ray diffraction data were measured using a Bruker D8 Venture dual-source single crystal X-ray diffractometer, employing Cu K α radiation ($\lambda=1.54178 \text{ \AA}$). Data were collected using a Photon II CMOS area detector on samples superglued to glass fibers. Reflection data were processed using the Bruker Apex 3 Suite, including an absorption correction through SADABS, and space group determination using XPREP. The structure was determined with SHELX-2019 software. The structure was solved with XT using intrinsic phasing and refined using full-matrix least-squares on F2 through XL. The structure was finalized in Olex 2 (v.1.3.0).*

Reviewer 2:

No quantum yield measurements have been carried out to quantify the efficiency of the compounds with each other? The authors could justify the choice not to use this quantitative method to study luminescence mechanisms that occur in these compounds.

The lifetime measurements have been measured several times for each compound? The values could be indicated with uncertainty errors in the tables. The authors should at least specify whether the repeatability of the measurement has been verified.

Response:

We thank the reviewer for bringing these important topics up for discussion. To clarify, this work's primary focus was placed on demonstrating an efficient and controllable strategy towards manipulating fluorescence lifetime, while the efficiency or brightness associated with these compounds was a secondary focus.

According to the reviewer's suggestion, quantum yield measurements were determined for two representative compositions, compounds **1** (Eu) and **13** (equimolar EuGdYb), under 340 nm and 394 nm excitation. These representative samples were chosen to highlight the effect of efficient ligand to metal transfer in the monometallic composition, vs. less efficient transfer in trimetallic compositions. Under 340 nm excitation, the spectral range of the emitted light detector extended out to 1100 nm, so the visible and near-IR emissions were detected. As the Eu concentration

decreased in compound **13**, the visible emission decreases, and the Eu to Yb energy transfer resulted in a much lower QY. This data has been added to the SI as Table S4.

Table S5.

Sample	340 nm	394 nm
Compound 1	5.5%	14.4%
Compound 13	1.1%	N/A

The methods section has been updated with details on the QY measurements.

Absolute Quantum Yield (QY) Measurements. QY was determined by exciting the sample with diffuse light inside an integrating sphere.⁴⁶ The powder samples were placed in Pyrex NMR tubes for these measurements and inserted into the integrating sphere. Both the diffuse excitation and emission light power were simultaneously recorded. The QYs were calculated from these power measurements.

46. Rohwer, L. S. & Martin, J. E. Measuring the absolute quantum efficiency of luminescent materials. *Journal of Luminescence* **115**, 77-90 (2005).

[https://doi.org:https://doi.org/10.1016/j.jlumin.2005.01.013](https://doi.org/https://doi.org/10.1016/j.jlumin.2005.01.013)

Further, to demonstrate the reproducibility of these measurements, the lifetime on the compound with the highest light output (compound **1**) was measured five times:

#1 – 367.06 μ s
#2 – 366.66 μ s
#3 – 368.67 μ s
#4 – 368.59 μ s
#5 – 367.56 μ s

The average lifetime for compound **1** is 367.71 μ s, with a standard deviation of +/-0.22%.

Importantly, it is more difficult to fit decay curves for samples with low light output because of the higher signal to noise level. In these circumstances, accurately correcting for the baseline is critical and subject to error. However, the overall lifetime measurements uncertainty for all compounds in this study is +/-5%, based on the equipment used. The methods section and the caption of Table 2 have been updated to include the following: “*The uncertainty of the measurements reported is +/- 5%.*”

Reviewer 2:

Is the thermal stability of these compounds known? The authors should recall this property if it has been investigated in previous studies or do an ATG-ATD or DSC analysis to obtain this information. For EDS analysis, the uncertainties of the measurements should be given in Table 1 (repeatability of the measures)

Response

We thank the reviewer for their comment and we appreciate the opportunity to address the materials stability. The materials reported herein are highly robust as inferred by their three-dimensional structures derived from polynuclear clusters. It has been previously shown that MOFs based on polynuclear clusters of metals with 3⁺ and 4⁺ oxidation states display superior thermal and chemical stabilities as compared to the vast majority of MOFs. These compounds are stable in air, in common organic solvents and water and are thermally stable up to 500°C. Thermogravimetric analyses (TGA) were conducted of select representative compounds and the results were included in the SI as Figure S1. To highlight the exceptional thermal stability in these compounds, in the revised version of the manuscript we have included the following text:

“Notably, these materials possess a high thermal stability up to 500°C, as determined via thermogravimetric analyses, Figure S1.”

Thermogravimetric analyses (TGA). Measurements were conducted on a SDTQ600 TA instrument. The samples were heated to 900°C at a 5 °C /min heating rate, under continuous nitrogen flow.

Figure S1. Thermogravimetric analyses (TGA) for two representative compounds, compounds **1** and **13**, highlighting the high thermal stability intrinsic to these materials.

To address the EDS concern, we have added an uncertainty to the fractional compositions based on repeatability to the caption of Table 1: “The reported fractional compositions have an uncertainty of +/- 0.01 based on EDS analysis.”

Reviewer 2:

The emission spectra in Figure 2 for europium and ytterbium were recorded at 395 nm (f-f band of Eu(III)), why not record these spectra under ligand excitation at 337 nm (antenna excitation). Moreover, in this perspective, it would be nice to add the excitation spectrum or the absorption spectrum of at least one compound justifying the 337 nm value for the TCPB absorption. Moreover, the authors give 26500 cm⁻¹ as the value of the TCPB ligand, the phosphorescence spectrum allowing this value to be determined should be provided in SI. In the figure 2.b, the value of the singlet is not given while that of the triplet is given at 25200 cm⁻¹. The authors should clarify this point. What does the value at 26500 cm⁻¹ correspond to? to the singlet, to the triplet? other assignment?

Furthermore, the Eu emission spectra show the most intense band at 617 nm, so why measure the lifetimes at 620 nm? Is this an instrumental choice related to the measurement? In the case of lifetimes, the presence of mathematical fits would allow a better understanding of the agreement between the experimental points and the fit model.

Response:

We thank the reviewer for their comments, and will address them point by point. The emission spectra in Figure 2 were measured using 395 nm excitation in order to highlight the peaks associated with Eu and Yb. These are present in the 337 nm excitation measurements but are dwarfed by the linker emission. Per the reviewer's suggestion, we have included an example of the 337 nm excitation spectra to the SI, in a new figure, Figure S4, for the TCPB linker and compounds **1**, **4** and **7**, measured at 400 nm to monitor linker excitation. Additionally, we have included the corresponding photoluminescent emission spectra for the TCPB linker and each of the single metal compounds, compounds **1**, **4** and **7**.

To highlight all these changes, the following text was included in the revised manuscript:

*“Additional excitation and emission spectra for the TCPB linker and single metal compounds **1**, **4** and **7** are presented in Figure S4.”*

Figure S4. a. Photoluminescent excitation spectra for the TCPB linker and compounds **1**, **4** and **7**, measured at 400 nm to monitor linker excitation; b. corresponding photoluminescent emission spectra for the TCPB linker and each of the single metal compounds, compounds **1**, **4** and **7**.

We are grateful to the reviewer's pointing out the triplet value discrepancy. This was a typo, the value of the triplet has been corrected to 25200 cm^{-1} .

The Eu lifetime was measured using a short pass filter centered at 620 nm, but with a full width at half max of 10 nm, so the entire peak was captured.

The methods section has been updated to reflect this: "A 620 nm narrow bandpass filter with FWHM of 10 nm was placed in front of the detector."

Reviewer 2:

In the method section, it would be good to give more details on the photoluminescence measurements performed on the fluorolog-3. What sources are used, what are the detectors of the device?

Response:

We appreciate the reviewer bringing this topic to our attention. A clarification has been added to the Fluorolog-3 details in the methods section: *“This instrument uses a continuous wave 450W Xenon arc lamp excitation source and a room temperature R928P photomultiplier tube (PMT) for the detection of visible light. A complete instrumental correction was performed on all spectra which compensates for factors such as the grating efficiencies and the wavelength-dependent PMT response.”*

Reviewer 3:

In the MS, the authors demonstrate a design strategy towards multiplexed, lifetime-encoded tags via engineering intermetallic energy transfer in heterometallic metal-organic frameworks. Compare to anticounterfeiting systems based on energy transfer between ligand and metal or fluorescence behavior of a single component, using excited lifetime for anticounterfeiting seems more complex, especially based on intermetallic energy transfer and heavily depend on the control over metal ordering. However, this referee thinks the design of this work is interesting and refreshing. I'd like to recommend the authors to make some revisions before publication.

Response:

We are pleased with the encouraging assessment of this manuscript.

Reviewer 3:

Some of the introduction parts are overstated, for example, not all amorphous physical structures are difficult to characterize, some pure organic compound with clear structure have been used for anticounterfeiting.

Response:

We appreciate the reviewer's comment. Our intention was to highlight that the inherent structure of MOFs makes them more conducive to fine tuning than many organic systems, however, in response to reviewer's concerned we have revised the content in specific areas of the introduction where this topic was discussed.

Reviewer 3:

Monitoring fluorescence of visible and near infrared regions simultaneously seems have higher requirements for equipment. And monitoring luminescence lifetime difference in microsecond seem difficult to popularize in practical application. The author should explain the advantage of this method.

Response:

We thank the reviewer for their comment. We agree that monitoring in both regimes simultaneously would be difficult, however we are suggesting that measurement separately would be fairly simple, requiring only different detectors, while still proving both overt and covert, difficult to counterfeit features.

Although microsecond regime lifetimes are difficult to detect with the naked eye or many conventional cameras, this technology is well within the capability of only slightly specialized equipment, given that even many phone cameras can approach speeds of 1000 fps (1000 μ s per frame). Beyond this, we suggest that the requirement for some level of specialized equipment would improve security over tags easily distinguished by the naked eye, while avoiding the incredibly specialized equipment required to measure lifetimes on the nanosecond scale, as displayed by the vast majority of technologies currently under development based on perovskite and carbon dot materials.

Reviewer 3:

Some closely related reference should be added: Nat. Commun. 2021, 12, 1363; Angew. Chem. Int. Ed. 2019, 58, 18025.

Response:

We thank the reviewer for pointing out these recent sources we had not initially cited. The text has been updated with references to these specific works (references 21 and 22), as noted below:

21. Li, Z. *et al.* Photoresponsive supramolecular coordination polyelectrolyte as smart anticounterfeiting inks. *Nature Communications* **12**, 1363, doi:10.1038/s41467-021-21677-4 (2021).
22. Li, Z. *et al.* Loading Photochromic Molecules into a Luminescent Metal–Organic Framework for Information Anticounterfeiting. *Angewandte Chemie International Edition* **58**, 18025-18031, doi:https://doi.org/10.1002/anie.201910467 (2019).

Reviewer 4:

In this work, the authors developed multiplexed, lifetime-encoded tags via engineering intermetallic energy transfer in heterometallic metal-organic frameworks. Manipulation of the luminescence decay dynamics over a wide microsecond regime was conducted by controlling the metal ordering in these systems. They then applied them as tags by incorporation into photocurable inks patterned on glass and interrogated via digital high-speed imaging. This work shows some important results to the field. Thus, it is recommended for publication after addressing the following issues.

Response:

We thank the reviewer for their time and input on this work and for the positive outlook on our study. We have addressed the concerns expressed as outlined below.

Reviewer 4:

It is better to give the chemical structures and formula of the heterometallic MOFs in the Supplementary Information for better clarity.

Response:

We appreciate the reviewer's suggestion. This aspect has been thoroughly addressed in our response to Reviewer 2, question 1.

Reviewer 4:

In addition to controlling the metal ordering in MOFs, is it possible to tune the organic ligands to achieve the targets as well?

Response:

We are grateful for the opportunity to clarify this point. As we state in the manuscript, the energy transfer relationships between elements of the material is a function of their excited state energies and the distance between them. It is most definitely possible to tune the properties of the materials using the organic ligand, primarily in how it governs intermetallic distances by directing the structure formation, as well as by acting as an energy absorber and donor for excitation. If the triplet state of the ligand were below the emissive state of Eu for example, Eu would act as Gd does in this manuscript, as a silent ion diluting the others and affecting their relationships. Given the structure directing nature of the ligands and their different energy levels, it would likely be a more complex prospect to achieve similar results using the ligand alone for any given metal composition.

Reviewer 4:

There are basically single-color tags. It will be better to achieve color-tunable tags for more advanced usage.

Response:

We thank the reviewer for their comment. We acknowledge that these are currently single-color tags, based solely on Eu emissions in the visible range. We chose this specifically to enable probing the lifetimes of the materials in a straightforward way and to constrain the number of different metals involved. The incorporation of additional visible emitters could be accomplished relatively easily to modify the tag color, as shown in other studies. To further outline the unique design prospects in this materials systems, in the revised version of the manuscript we have included the following paragraph: *“While not demonstrated here, additional layers of encoding could be introduced either through the presence of different emitting REs or their ratios (i.e. differentiating the encoding via spectra or color), through the presence of inactive REs like Gd that are detectable via X-ray fluorescence or EDS (i.e. differentiating by composition independent of the measurable spectra), or through the inclusion of additional molecules in the MOF pore-space.”*

Reviewer 4:

Computational studies should be performed to support the energy transfer within the selected systems.

Response:

We thank the reviewer for their suggestion. As recommended, in the revised version of the manuscript we have included density functional theory (DFT) electronic structures calculations. Projected density of states (PDOS) calculations are well suited to provide individual elemental contributions to the overall homometallic and heterometallic cluster electronic structures, thus helping elucidate the complex correlation between metallic compositions and resulting unique photophysical signatures. The results of the calculated electronic structures support two primary trends identified in the experimentally observed spectra: i) increased Eu content results in longer

luminescence lifetimes and ii) increased Gd content disrupts intermetallic energy transfer between Eu and Yb.

We have included an extensive discussion on these results both in the main manuscript text, as well as supporting information, as detailed below.

*To gather additional insights into how heterometallic compositions and relative atomic distribution impact the electronic structure of the materials, further investigation has been conducted using density functional theory (DFT) calculations. Projected density of states (PDOS) calculations are well suited to provide individual elemental contributions to the overall homometallic and heterometallic cluster electronic structures, thus helping elucidate the complex correlation between metallic compositions and resulting unique photophysical signatures. Accordingly, heterometallic clusters were modeled for the six compounds highlighted in **Figure 5a**, along with the three homometallic clusters, **Figure S10-S11**, to identify qualitative information in correlating decay times with metallic distributions. For simplicity, the compounds in **Figure 5a** can be categorized by decay rate as fast (compounds **3** and **11**), intermediate (compounds **13** and **2**), and slow (compounds **12** and **10**). Models for compounds **3**, **13**, and **12** are presented as exemplars for the three luminescence lifetimes regimes to highlight the impact of heterometallic distribution, **Figure 5b**.*

*The calculated electronic structures of all metallic clusters identify two primary peaks near the edge of the conduction band and are clearly marked in **Figure 5b**. To emphasize the correlation of metallic distribution with electronic structure, in **Figure 5b** are presented the RE only PDOS for clarity. Notably, the peaks contain contributions from all elements within the cluster, with large densities localized on C and O atoms, **Figure S11**. The hybridization of the organic components with the RE species is expected as the metallic clusters are bridged via organic linkers in the periodic material.*

*The results of the calculated electronic structures support two primary trends identified in the experimentally observed spectra: i) increased Eu content results in longer luminescence lifetimes and ii) increased Gd content disrupts intermetallic energy transfer between Eu and Yb. The calculated PDOS for models shows a matching trend of increasing Eu:Yb density at the first conduction band peak and models from fastest to slowest luminescence decay, **Figure 5** and **Figure S10**. The higher PDOS of Eu at the lower energy state indicates that longer lived charge carriers nonradiatively migrate to Eu prior to any optical emission event, supporting the observed experimental results.*

*The introduction of Gd into the heterometallic clusters is hypothesized to impact the optical response by increasing spatial separation between the optically active Eu and Yb species in the individual clusters. Spatial separation reduces the transition dipole moment between Eu and Yb species, reducing the likelihood of the event. Electronic structure calculations reinforce this premise revealing that introduction of Gd into the heterometallic gives rise to new electronic states. Importantly, the presence of Gd directly impacts the increased Eu:Yb ratio at the first conduction band peak, **Figure 5b**, further supporting long lived charge carriers nonradiatively relaxing to Eu and resulting in long lived photoluminescence lifetimes.*

Figure 5.b. Calculated RE projected density of states (PDOS) near the electronic band edges for three representative heterometallic clusters 3Eu-6Yb (left), 3Eu-3Gd-3Yb (center), and 1Eu-7Gd-1Yb (right). The PDOS identify the relative electron density localized on each of the three RE elements; Eu (red), Yb (blue), and Gd (green). Each panel highlights the two primary peaks of RE electron density in the conduction band (black dashed box) which participate in excited state relaxation mechanisms. The three panels are organized by relative luminescence decay time as measured in this experimental work.

*In the 6Yb-3Eu model, the first peak shows a combination of Yb and Eu contribution with the ratio of localized density on Eu:Yb as 2:1 while the second peak shows an inverted relationship. The proposed relaxation mechanisms, **Figure 3b**, indicate that fast decay rates in the heterometallic clusters are attributed to relaxation on the Yb atoms, which shows a higher localized density at the second peak. This second peak is at a higher energy and is closer to the initially excited energy states contributed by the organic linkers which would require less energy dissipation to transfer the charge to Yb prior to optical emission. Similar peak ordering is seen in the PDOS of the other fast decay rate compound, **11**. In the calculated PDOS, **Figure S10**, the second peak is dominated by Yb while the first peak has contributions from Gd>Eu>Yb even though the composition is dominated by Yb.*

Figure S10. a. Calculated rare earth projected density of states for three heterometallic clusters 6Yb-2Gd-1Eu (left), 6Eu-3Yb (center), and 6Eu-2Gd-1Yb (right). The PDOS identify the relative electron density localized on each of the three RE elements Eu (red), Yb (blue), and Gd (green).

For the intermediate decay rate, the equimolar 3Eu-3Gd-3Yb cluster, **Figure 5b**, increasing Eu and Gd concentrations modifies the relative amplitudes of electron localization at the first and second peaks. The first peak becomes delocalized onto all RE elements, heavily localizing onto Eu. The resulting relative ratios for the first peak are not the equimolar ratio of the cluster, highlighting a strong preference for Eu at the conduction band edge. It also indicates that Gd, while optically inactive, reduces the density localized on the Yb species.

The second peak of the equimolar cluster shows an electron density localization that nearly maintains the equimolar ratio of 3:3:3 for Eu:Yb:Gd with Gd being the least of the three RE species. This distribution of localized density demonstrates that both Eu and Yb are still participating in possible high energy states that may receive charge from the excited organic linkers. This relative participation hypothesis agrees with the observed relative photoluminescence lifetimes comparing the equimolar cluster to the 3Eu-6Yb cluster.

The second intermediate decay rate compound, **2**, has a composition of 6Eu-3Yb. The calculated PDOS for compound **2**, **Figure S10**, shows a near equivalent mixing of Eu and Yb at the first peak. This is unique in that for all other modeled systems the Eu dominates the first peak at the conduction band edge. However, the second peak is dominated by Eu at a rate of 2:1 for Eu:Yb.

The composition of the slowest decay rate, compound **12**, has a composition heavily dominated by Gd: 1Eu-7Gd-1Yb. The Gd expectedly dominates the electron density localization as it is the primary metal in the cluster. However, for the two optically active elements, Eu and Yb, the Eu is the stronger participant in the conduction band structure and would be expected to dominate any optically active relaxation pathways. While at the second peak the Yb does show an increase peak amplitude compared to Eu, the long lifetime is expected as the Eu localizes more charge than Yb

and there is substantial physical separation between the Eu and Yb elements in a Gd dominated cluster. The PDOS of the last slow decay rate compound, **10**, shows very minimal participation in the conduction band electronic structure by Yb, **Figure S10**. This is expected as there is only one Yb atom in the 6Eu-2Gd-1Yb cluster. At both the first and second conduction band peaks Eu dominates and Gd shows more participation than Yb. The small concentration of Yb does not have enough interaction with the Eu atoms or organic linker excited states to be optically active.

Figure S11. Left: Calculated RE PDOS for homometallic Eu (red), Gd (green), and Yb (blue) clusters plotted together for comparison. Right: Full elemental PDOS for homometallic Eu cluster. The total elemental contributions to the PDOS (solid black) are identified for Eu (solid red), C (dashed cyan), O (dashed magenta), and H (dashed green). The homometallic Eu cluster was chosen as a representative exemplar for the series of modeled clusters.

Finally, the methods section has been updated with details on the computational models and methodology and the references section with references 47 through 60.

Computational Models and Methodology

The geometry optimization and ground state electronic structures of the metallic clusters were calculated using spin-restricted density functional theory (DFT) as implemented in the Vienna *ab initio* simulation package (VASP).^{47,48} The simulated cluster models were built with the following molecular formula: $[M_9(\text{HCO}_2)_{12}(\mu_3\text{OH})_{12}\text{O}_2]^- \text{Na}^+$ where $M = \text{Eu}, \text{Yb}, \text{Gd}$ and a non-interactive Na atom is included in the simulation cell to balance the total charge of the simulation cell.

The calculations were completed in a plane wave basis set^{49,50} with a 500eV cutoff energy and converged to a force accuracy of 0.01eV/Å at the gamma point. The generalized gradient approximation (GGA) exchange-correlation functional of Perdew, Burke, and Ernzerhof designed for solids and surfaces (PBEsol)⁵¹ was used due to previous calculation sets.^{52,53,54,55,56} The DFT-D3 method of Grimme *et al.*⁵⁷ With Becke-Johnson damping⁵⁸ was used for dispersion corrections. Projector-augmented wave (PAW) potentials^{59,60} were used for C, O, H, and Na, with the large core potentials (LCPs) which represent the M(III) oxidation state used for Eu, Yb, and Gd. These potentials place the 4f electrons in the core of the potential. This approximation was utilized as an initial set of calculations to investigate the changes in core electronic structure when complex

combinations of rare earth metals are combined. The calculation set provides information on electronic interactions prior to charge being transferred to the 4f orbitals of the Eu, Yb, and Gd. The projected density of states (PDOS) were calculated according to the VASP protocol using LORBIT=11. This provides spatially resolved electron density as a function of spherical volume at each atomic location. The PDOS calculations utilized the standard RWIGS values provided in each elements PBE POTCAR file. While spherical limits on electron density are known to underpredict hybridized electrons in space, the primary atomic contributions calculated in the PDOS were confirmed by visualizing molecular orbitals from calculated PARCHG files which are derived from the calculated wavefunction.

- 47 Kresse, G. & Hafner, J. Ab initio molecular dynamics for liquid metals. *Physical Review B* **47**, 558-561 (1993). <https://doi.org/10.1103/PhysRevB.47.558>
- 48 Kresse, G. & Hafner, J. Ab initio molecular-dynamics simulation of the liquid-metal--amorphous-semiconductor transition in germanium. *Physical Review B* **49**, 14251-14269 (1994). <https://doi.org/10.1103/PhysRevB.49.14251>
- 49 Kresse, G. & Furthmüller, J. Efficiency of ab-initio total energy calculations for metals and semiconductors using a plane-wave basis set. *Computational Materials Science* **6**, 15-50 (1996). [https://doi.org:https://doi.org/10.1016/0927-0256\(96\)00008-0](https://doi.org/https://doi.org/10.1016/0927-0256(96)00008-0)
- 50 Kresse, G. & Furthmüller, J. Efficient iterative schemes for ab initio total-energy calculations using a plane-wave basis set. *Physical Review B* **54**, 11169-11186 (1996). <https://doi.org/10.1103/PhysRevB.54.11169>
- 51 Perdew, J. P. *et al.* Restoring the Density-Gradient Expansion for Exchange in Solids and Surfaces. *Physical Review Letters* **100**, 136406 (2008). <https://doi.org/10.1103/PhysRevLett.100.136406>
- 52 Sava Gallis, D. F., Vogel, D. J., Vincent, G. A., Rimsza, J. M. & Nenoff, T. M. NO_x Adsorption and Optical Detection in Rare Earth Metal–Organic Frameworks. *ACS Applied Materials & Interfaces* **11**, 43270-43277 (2019). <https://doi.org/10.1021/acsami.9b16470>
- 53 Vogel, D. J., Sava Gallis, D. F., Nenoff, T. M. & Rimsza, J. M. Structure and electronic properties of rare earth DOBDC metal–organic–frameworks. *Physical Chemistry Chemical Physics* **21**, 23085-23093 (2019). <https://doi.org/10.1039/C9CP04038B>
- 54 Vogel, D. J., Nenoff, T. M. & Rimsza, J. M. Tuned Hydrogen Bonding in Rare-Earth Metal–Organic Frameworks for Design of Optical and Electronic Properties: An Exemplar Study of Y–2,5-Dihydroxyterephthalic Acid. *ACS Applied Materials & Interfaces* **12**, 4531-4539 (2020). <https://doi.org/10.1021/acsami.9b20513>
- 55 Henkelis, S. E. *et al.* Kinetically Controlled Linker Binding in Rare Earth-2,5-Dihydroxyterephthalic Acid Metal–Organic Frameworks and Its Predicted Effects on Acid Gas Adsorption. *ACS Applied Materials & Interfaces* **13**, 56337-56347 (2021). <https://doi.org/10.1021/acsami.1c17670>
- 56 Henkelis, S. E., Huber, D. L., Vogel, D. J., Rimsza, J. M. & Nenoff, T. M. Magnetic Tunability in RE-DOBDC MOFs via NO_x Acid Gas Adsorption. *ACS Applied Materials & Interfaces* **12**, 19504-19510 (2020). <https://doi.org/10.1021/acsami.0c01813>
- 57 Grimme, S., Antony, J., Ehrlich, S. & Krieg, H. A consistent and accurate ab initio parametrization of density functional dispersion correction (DFT-D) for the 94 elements

- H-Pu. *The Journal of Chemical Physics* **132**, 154104 (2010).
<https://doi.org/10.1063/1.3382344>
- 58 Grimme, S., Ehrlich, S. & Goerigk, L. Effect of the damping function in dispersion corrected density functional theory. *Journal of Computational Chemistry* **32**, 1456-1465 (2011). <https://doi.org/10.1002/jcc.21759>
- 59 Blöchl, P. E. Projector augmented-wave method. *Physical Review B* **50**, 17953-17979 (1994). <https://doi.org/10.1103/PhysRevB.50.17953>
- 60 Kresse, G. & Joubert, D. From ultrasoft pseudopotentials to the projector augmented-wave method. *Physical Review B* **59**, 1758-1775 (1999).
<https://doi.org/10.1103/PhysRevB.59.1>

Reviewer comments, second round –

Reviewer #1 (Remarks to the Author):

The authors revised the paper according to my previous suggestions. And it can be published as its current form.

Reviewer #2 (Remarks to the Author):

The revisited manuscript by Deneff, Rohwer, Butler, Kaehr, Vogel, Luk, Reyes, Cruz-Cabrera, Martin, Sava Gallis present the synthesis, structural and optical characterizations of REMOF or LnTCPB compounds. This work shows a fundamental study of the luminescence mechanisms from emission spectra but especially from lifetime measurements. Additional DFT calculations have been added by the authors which answers the question of one of the referees and greatly improves the quality of the manuscript. The authors have taken the time to answer all the questions of the referees. Moreover, they also performed additional manipulations sometimes on their own initiative to meet the expectations of the referees. For all these reasons, the manuscript should now be accepted in Nature Communications.

Reviewer #3 (Remarks to the Author):

All of my questions have been addressed, I recommend its publication in Nature Communications

Reviewer #4 (Remarks to the Author):

I am satisfactory for the revision. The revised manuscript is recommended for publication.

REVIEWERS' COMMENTS

Reviewer 1:

The authors revised the paper according to my previous suggestions. And it can be published as its current form.

Reviewer 2:

The revisited manuscript by Deneff, Rohwer, Butler, Kaehr, Vogel, Luk, Reyes, Cruz-Cabrera, Martin, Sava Gallis present the synthesis, structural and optical characterizations of REMOF or LnTCPB compounds. This work shows a fundamental study of the luminescence mechanisms from emission spectra but especially from lifetime measurements. Additional DFT calculations have been added by the authors which answers the question of one of the referees and greatly improves the quality of the manuscript. The authors have taken the time to answer all the questions of the referees. Moreover, they also performed additional manipulations sometimes on their own initiative to meet the expectations of the referees. For all these reasons, the manuscript should now be accepted in Nature Communications.

Reviewer 3:

All of my questions have been addressed, I recommend its publication in Nature Communications

Reviewer 4:

I am satisfactory for the revision. The revised manuscript is recommended for publication.

Response:

We thank all the reviewers for their time, effort, critical read, and very helpful suggestions on this work and are very pleased to learn their unanimous final recommendation for publication.